# Quantization Hurts Reasoning? An Empirical Study on Quantized Reasoning Models

**Ruikang Liu**$^{\diamond *}$, **Yuxuan Sun**$^{\spadesuit *}$, **Manyi Zhang**$^{\spadesuit *}$, **Haoli Bai**$^{\spadesuit \#}$, **Xianzhi Yu**$^{\spadesuit}$,
**Tiezheng Yu**$^{\spadesuit}$, **Chun Yuan**$^{\diamond}$, **Lu Hou**$^{\spadesuit \#}$
$^{\diamond}$Shenzhen International Graduate School, Tsinghua University
$^{\spadesuit}$Huawei Noah's Ark Lab
{baihaoli,houlu3}@huawei.com

## Abstract

Recent advancements in reasoning language models have demonstrated remarkable performance in complex tasks, but their extended chain-of-thought reasoning process increases inference overhead. While quantization has been widely adopted to reduce the inference cost of large language models, its impact on reasoning models remains understudied. In this paper, we conduct the first systematic study on quantized reasoning models, evaluating the open-sourced DeepSeek-R1-Distilled Qwen and LLaMA families ranging from 1.5B to 70B parameters, QwQ-32B, and Qwen3-8B. Our investigation covers weight, KV cache, and activation quantization using state-of-the-art algorithms at varying bit-widths, with extensive evaluation across mathematical (AIME, MATH-500), scientific (GPQA), and programming (LiveCodeBench) reasoning benchmarks. Our findings reveal that while lossless quantization can be achieved with W8A8 or W4A16 quantization, lower bit-widths introduce significant accuracy risks. We further identify model size, model origin, and task difficulty as critical determinants of performance. Contrary to expectations, quantized models do not exhibit increased output lengths. In addition, strategically scaling the model sizes or reasoning steps can effectively enhance the performance. All quantized models and codes are open-sourced in https://github.com/ruikangliu/Quantized-Reasoning-Models.

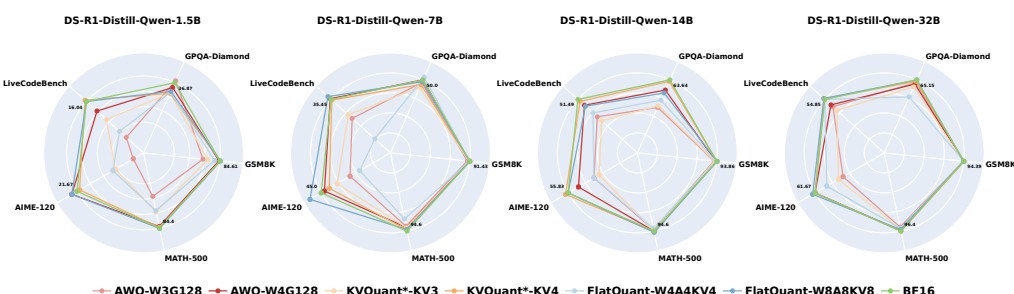

Figure 1: Performance of the quantized DeepSeek-R1-Distill-Qwen models on five benchmarks: AIME-120, MATH-500, GSM8K, GPQA-Diamond, and LiveCodeBench. Each chart presents the performance of different quantization strategies compared to the BF16 baseline.

## 1 Introduction

Recent large language models (LLMs) (Jaech et al., 2024; Team et al., 2025; Guo et al., 2025; Team, 2025) trained with enhanced reasoning abilities demonstrate strong performance on

---

$^{*}$Equal contribution; #: Corresponding authors.

complex reasoning tasks, such as multi-disciplinary question answering (Rein et al., 2024) and mathematical competitions (Maxwell-Jia, 2025). However, their improved performance increases inference overhead from prolonged reasoning processes, often causing them to "overthink" simple tasks and generate outputs hundreds of times longer than non-reasoning models (Chen et al., 2024).

To promote efficient LLM inference, various compression and acceleration techniques have been proposed, including quantization (Bai et al., 2021; Frantar et al., 2022; Xiao et al., 2023; Liu et al., 2024), pruning (Frantar & Alistarh, 2023; Sun et al.; Zhang et al., 2024; Chen et al., 2025), and knowledge distillation (Gu et al., 2023; Liu et al., 2023; Muralidharan et al., 2024). Among these methods, quantization, which reduces the numerical precision of weights or activations, is particularly favored due to the good balance between model size and performance preservation. Nonetheless, most existing quantization methods are proposed for non-reasoning LLMs (Li et al., 2025), and their performance on reasoning models remains under-explored. Intuitively, reasoning models are at higher risks of performance degradation by quantization, since the quantization error may accumulate along the long chain-of-thought (CoT) reasoning steps (Wei et al., 2022).

In this study, we provide a comprehensive empirical study on the quantization of reasoning models, as outlined in Figure 1. We evaluate state-of-the-art quantization methods across weight-only, weight-activation, and KV cache quantization. The reasoning models for evaluation span form 1.5B to 70B parameters, such as the series of the open-source DeepSeek-R1-Qwen-Distill (Yang et al., 2024; Guo et al., 2025), DeepSeek-R1-LLaMA-Distill (AI@Meta, 2024; Guo et al., 2025), QwQ (Team, 2025), and Qwen3 (Yang et al., 2025), with prevalent reasoning benchmarks such as AIME-120 (Maxwell-Jia, 2025), MATH-500 (Lightman et al., 2023), GSM8K (Cobbe et al., 2021). GPQA-Diamond (Rein et al., 2024) and Live-CodeBench (Jain et al., 2024). The key findings of this study are summarized below:

1. **Lossless Quantization (§3.2)** : 8-bit weight-activation quantization preserves accuracy across tasks and model sizes, while 4-bit weight-only quantization or KV cache quantization also achieves near-lossless results ($\leq$1% drop).

2. **Quantization Algorithm (§3.3)** : Among the evaluated quantization algorithms, we suggest AWQ for weight-only quantization and QuaRot for KV cache quantization. While SmoothQuant suffices for 8-bit weight-activation quantization, FlatQuant takes the lead in the 4-bit scenario.

3. **Impact of Task Difficulty (§3.4)** : Harder tasks (e.g., AIME-120) suffer up to $4\times$ greater degradation than simpler ones (e.g., GSM8K).

4. **Model Origins Matter (§3.5)** : Distillation-based and RL-based reasoning models display varying levels of tolerance to quantization, even when derived from the same base model. Additionally, different model families (e.g., Qwen and LLaMA) exhibit distinct sensitivities to quantization.

5. **Analysis of Output Length (§4.1)** : Quantized models with minor performance drops do not exhibit longer reasoning steps, but aggressive low-bit quantization can lead to increased output lengths, particularly in smaller models.

6. **Scaling Effects (§4.3 and §4.4)** : Larger quantized models achieve superior accuracy-latency trade-offs compared to smaller BF16 models, and while longer reasoning steps improve performance, the gains diminish at lower rates than in BF16 models.

This work is among the first attempts to comprehensively evaluate the quantization of reasoning models, concurrent to Kurtić et al. (2025). We hope our research provides valuable guidance for the community toward better quantization methods for reasoning models.

## 2 Preliminary and Related Work

### 2.1 Quantization

Quantization converts high-precision values in $\mathbf{X} \in \mathbb{R}^{m \times n}$ to lower-precision representations. In this study, we focus mainly on hardware-efficient uniform quantization, where the $b$-bit

quantization function $\mathcal{Q}(\,\cdot\,;b)$ can be represented by

$$\hat{\mathbf{X}} = \mathcal{Q}(\mathbf{X};b) = s \cdot \Pi_{\Omega(b)}(\mathbf{X}/s), \tag{1}$$

where $s = \frac{\max(\mathbf{X})-\min(\mathbf{X})}{2^b-1} \in \mathbb{R}^+$ is the quantization step size, $\Pi(\cdot)$ is the projection function that maps the value to the closest element in the set $\Omega(b) = \{0, 1, ..., 2^b - 1\}$ of $b$-bit integer points. The scope of this study includes the following quantization configurations, where more implementation details of these quantization algorithms can be found in Appendix A.

**Weight-only Quantization.** Following the conventional practice (Frantar et al., 2022), we quantize the weight matrices of all linear layers in the Transformer. and leave the rest parameters in BF16. By quantizing the values of weight matrix $\mathbf{W}$ into low-bit integers, the model size and memory access can be effectively reduced during runtime. We focus mainly on GPTQ (Frantar et al., 2022) and AWQ (Lin et al., 2023), two widely applied approaches in the research community. We adopt asymmetric quantization with a group size of 128, i.e., the step size is calculated per 128 consecutive elements per channel in $\mathbf{W}$.

**KV Cache Quantization.** The KV cache temporarily stores intermediate results during autoregressive generation to improve computational efficiency. The quantization of the KV cache is particularly helpful in reducing the memory access in long sequences, i.e., reasoning with long chain-of-thought steps. We study the recently popular methods, including QuaRot (Ashkboos et al., 2024) and KVQuant (Hooper et al., 2024). For QuaRot, we adopt asymmetric per-token KV cache quantization with a group size of 128. We implement KVQuant* as a simplified baseline, with details in Appendix A.2. KVQuant* adopts static per-channel asymmetric quantization for K cache and per-token asymmetric quantization for V cache with a group size of 128.

**Weight-activation Quantization.** Weight-activation quantization employs low-precision arithmetic for matrix multiplications to further save computation. The evaluated algorithms include SmoothQuant (Xiao et al., 2023), QuaRot (Ashkboos et al., 2024), and FlatQuant (Sun et al., 2024). Following the convention of these methods, we apply per-channel symmetric quantization to weights and per-token asymmetric quantization to activations. We quantize all linear layers in the Transformer, including the KV cache. The KV cache is quantized using group-wise asymmetric quantization with a group size of 128. We leave the queries in full precision, following Ashkboos et al. (2024). Additionally, inspired by the recent progress on FP4 inference (NVIDIA, 2025), we evaluate MXFP4 (Rouhani et al., 2023) quantization, where every contiguous 32 elements share the same scale.

## 2.2 Related Work

**Families of Reasoning Models.** Recent advancements in reasoning models like OpenAI's o1 (Jaech et al., 2024) and DeepSeek's R1 (Guo et al., 2025) have demonstrated the potential to solve complex tasks that require logical deduction, multi-step reasoning, and contextual understanding. These models can be broadly categorized based on how they achieve their reasoning abilities. The first category of reasoning models achieves the reasoning ability by distilling from the outputs of existing stronger models. For instance, DeepSeek-R1-Distill-Qwen and DeepSeek-R1-Distill-LLaMA are distilled from 800K curated samples from DeepSeek-R1 (Guo et al., 2025). The other type of reasoning models obtain the reasoning ability via self post-training techniques like supervised finetuning (Ye et al., 2025; Muennighoff et al., 2025), reinforcement learning (RL) (Guo et al., 2025; Team, 2025), or their combinations (Team, 2025; Team et al., 2025; Guo et al., 2025).

**Efficient Reasoning.** Despite the enhanced reasoning abilities, the long chain-of-thought reasoning process also incurs a greater challenge of inference overhead due to the increased output length. To reduce the inference cost, several approaches have been proposed. For reasoning models trained with reinforcement learning, a common way is to add a length penalty term on the reward in addition to the accuracy (Luo et al., 2025; Aggarwal & Welleck, 2025; Team et al., 2025). Supervised fine-tuning with variable-length Chain-of-Thought

(CoT) data is another effective approach. For instance, Token-Budget (Han et al., 2024) guides LLMs to complete reasoning within a specified token limit, while S1 (Muennighoff et al., 2025) controls the output length through forcefully terminating the model's thinking process or lengthening it by appending "Wait" multiple times. One can also reduce the reasoning lengths via direct performance optimization (DPO), where short and long CoT data can be curated as positive and negative data pairs (Chen et al., 2024; Team et al., 2025), or merging the long CoT model with a short CoT model (Team et al., 2025). Apart from training, efficient reasoning can also be realized by better test-time scaling with parallel inference (Rodionov et al., 2025; Wang et al., 2025; Pan et al., 2025).

Among the above efforts, quantization is an orthogonal method to achieve efficient reasoning. We believe a comprehensive empirical study on quantized reasoning models is essential to fully understand the trade-offs between reasoning efficiency and accuracy.

## 3 Evaluation of Quantized Reasoning Models

### 3.1 Setup

**Evaluation Benchmarks.** We evaluate the quantized models with the algorithms in § 2.1 on the following reasoning benchmarks. 1) Three mathematical reasoning benchmarks sorted by their difficulty: AIME-120 which consists of 120 problems from the American Invitational Mathematics Examination (AIME) from 2022 to 2025 to minimize evaluation variations; MATH-500 (Lightman et al., 2023), a benchmark that contains a mix of easy and hard mathematical problems designed to test comprehensive reasoning abilities; and GSM8K (Cobbe et al., 2021), a dataset composed of primary school level questions focused on basic arithmetic and algebra. 2) LiveCodeBench (Jain et al., 2024), a benchmark for evaluating large language models on code generation tasks, designed to assess their ability to produce high-quality and functional code. 3) GPQA-Diamond (Rein et al., 2024), a graduate-level proof question and answer benchmark that tests the ability of models to generate accurate mathematical proofs. We use Lighteval (Fourrier et al., 2023) with the vLLM (Kwon et al., 2023) backend for evaluation, with a sampling temperature of 0.6 and top-$p$ of 0.95. The maximum number of generation tokens is limited to 32,768. For all results, we repeat the experiments with three different seeds to reduce the evaluation variations.

**The Evaluated Reasoning Models.** We evaluate the two categories of reasoning models as described in § 2.2. For distillation-based reasoning models, we adopt the series of DeepSeek-R1-Distill-Qwen (Yang et al., 2024) with varying sizes from 1.5B, 7B, 14B and 32B, and DeepSeek-R1-Distill-LLaMA (AI@Meta, 2024), with both 8B and 70B models. For reasoning models obtained via reinforcement learning, we choose QwQ-32B (Team, 2025) and Qwen-3-8B (Yang et al., 2025), the recent performant open-source models.

We study the following research questions in the next sections.

> **RQ1** (§3.2): What are the **lossless quantization configurations** for reasoning models?
>
> **RQ2** (§3.3): What are the preferred **quantization algorithms** on reasoning tasks?
>
> **RQ3** (§3.4): How does the **task difficulty** influence the quantized LLMs?
>
> **RQ4** (§3.5): What is the impact of **LLM origins** on quantization?

### 3.2 Lossless Quantization Bit-width

The overall results for the DeepSeek-R1-Distill-Qwen models are summarized in Table 1. Results on more models (including DeepSeek-R1-Distill-LLaMA, QwQ-32B, and Qwen3-8B) can be found in Appendix B.1. For ease of interpretation, we categorize the performance degradation into three classes: lossless ($\leq 1\%$), fair ($1\% - 3\%$), and risky ($\geq 3\%$). We draw the following conclusions based on best-performing quantization algorithms, and leave the algorithm comparisons in § 3.3.

| Model | Quantization | W-A-KV # Bits | Methods | AIME-120 | MATH-500 | GSM8K | GPQA-Diamond | LiveCode-Bench | Avg. | Drop↓ |
|---|---|---|---|---|---|---|---|---|---|---|
| DeepSeek-R1-Distill-Qwen-1.5B | **BF16** | - | - | $23.3_{\pm2.2}$ | $84.7_{\pm1.5}$ | $84.5_{\pm1.3}$ | $36.2_{\pm0.8}$ | $16.4_{\pm1.1}$ | $49.0_{\pm0.3}$ | - |
| | *Weight-only* | 4-16-16 | AWQ | $21.1_{\pm2.1}$ | $81.4_{\pm1.6}$ | $82.7_{\pm0.7}$ | $35.4_{\pm2.7}$ | $13.9_{\pm1.5}$ | $46.9_{\pm1.2}$ | -2.1 |
| | | | GPTQ | $21.4_{\pm3.9}$ | $83.0_{\pm1.0}$ | $83.3_{\pm0.6}$ | $32.0_{\pm5.2}$ | $13.4_{\pm0.8}$ | $46.6_{\pm1.9}$ | -2.4 |
| | | 3-16-16 | AWQ | $4.4_{\pm1.3}$ | $48.1_{\pm0.3}$ | $64.5_{\pm1.3}$ | $31.8_{\pm6.1}$ | $3.2_{\pm1.1}$ | $30.4_{\pm1.3}$ | -18.6 |
| | | | GPTQ | $15.6_{\pm1.3}$ | $75.3_{\pm0.4}$ | $77.8_{\pm0.7}$ | $29.8_{\pm4.0}$ | $8.7_{\pm0.8}$ | $41.4_{\pm0.7}$ | -7.6 |
| | *KV Cache* | 16-16-4 | KVQuant* | $20.3_{\pm1.0}$ | $83.9_{\pm0.3}$ | $84.2_{\pm0.6}$ | $34.0_{\pm2.8}$ | $15.9_{\pm1.4}$ | $47.7_{\pm0.4}$ | -1.4 |
| | | | QuaRot | $0.3_{\pm0.5}$ | $1.5_{\pm0.1}$ | $0.9_{\pm0.2}$ | $9.6_{\pm2.0}$ | $0.0_{\pm0.0}$ | $2.5_{\pm0.5}$ | -46.6 |
| | | 16-16-3 | KVQuant* | $6.7_{\pm2.5}$ | $65.7_{\pm1.2}$ | $70.2_{\pm0.7}$ | $30.5_{\pm1.1}$ | $10.8_{\pm0.4}$ | $36.8_{\pm0.9}$ | -12.3 |
| | | | QuaRot | $0.0_{\pm0.0}$ | $1.3_{\pm0.4}$ | $0.9_{\pm0.3}$ | $21.0_{\pm2.8}$ | $0.0_{\pm0.0}$ | $4.7_{\pm0.6}$ | -44.4 |
| | *Weight-Act.* | 8-8-8 | SmoothQuant | $16.9_{\pm2.4}$ | $78.9_{\pm1.0}$ | $83.0_{\pm0.8}$ | $31.3_{\pm2.2}$ | $15.7_{\pm1.0}$ | $45.2_{\pm0.4}$ | -3.9 |
| | | | QuaRot | $22.8_{\pm1.3}$ | $84.2_{\pm0.9}$ | $83.8_{\pm0.8}$ | $38.4_{\pm2.2}$ | $15.7_{\pm2.3}$ | $49.0_{\pm1.0}$ | -0.1 |
| | | | FlatQuant | $22.5_{\pm0.8}$ | $85.0_{\pm0.5}$ | $84.7_{\pm0.4}$ | $37.0_{\pm4.1}$ | $16.8_{\pm0.4}$ | $49.2_{\pm0.6}$ | 0.2 |
| | | 4-4-4 | MXFP4 | $0.0_{\pm0.0}$ | $1.1_{\pm0.5}$ | $0.6_{\pm0.2}$ | $18.4_{\pm2.5}$ | $0.0_{\pm0.0}$ | $4.0_{\pm0.4}$ | -45.0 |
| | | | QuaRot | $0.0_{\pm0.0}$ | $1.4_{\pm1.1}$ | $0.9_{\pm0.3}$ | $12.0_{\pm1.1}$ | $0.0_{\pm0.0}$ | $2.9_{\pm0.2}$ | -46.2 |
| | | | FlatQuant | $8.6_{\pm1.3}$ | $66.1_{\pm1.7}$ | $76.8_{\pm2.0}$ | $32.0_{\pm0.8}$ | $7.5_{\pm0.8}$ | $38.2_{\pm0.3}$ | -10.9 |
| DeepSeek-R1-Distill-Qwen-7B | **BF16** | - | - | $46.1_{\pm1.0}$ | $93.9_{\pm0.7}$ | $91.2_{\pm0.6}$ | $51.0_{\pm1.0}$ | $36.7_{\pm2.5}$ | $63.8_{\pm0.6}$ | - |
| | *Weight-only* | 4-16-16 | AWQ | $42.2_{\pm1.0}$ | $92.5_{\pm1.4}$ | $90.7_{\pm0.4}$ | $48.1_{\pm2.8}$ | $35.1_{\pm2.9}$ | $61.7_{\pm1.1}$ | -2.1 |
| | | | GPTQ | $44.4_{\pm3.4}$ | $93.3_{\pm0.1}$ | $91.0_{\pm0.7}$ | $49.0_{\pm1.0}$ | $33.3_{\pm1.1}$ | $62.2_{\pm0.8}$ | -1.6 |
| | | 3-16-16 | AWQ | $32.2_{\pm3.2}$ | $91.0_{\pm1.4}$ | $89.5_{\pm0.4}$ | $47.8_{\pm1.2}$ | $27.2_{\pm1.0}$ | $57.6_{\pm0.8}$ | -6.3 |
| | | | GPTQ | $37.2_{\pm2.9}$ | $91.2_{\pm0.2}$ | $89.7_{\pm0.6}$ | $47.6_{\pm2.5}$ | $26.4_{\pm1.1}$ | $58.4_{\pm0.8}$ | -5.4 |
| | *KV Cache* | 16-16-4 | KVQuant* | $44.4_{\pm3.8}$ | $93.4_{\pm0.2}$ | $90.9_{\pm0.7}$ | $49.0_{\pm1.5}$ | $36.3_{\pm0.8}$ | $62.8_{\pm1.0}$ | -1.0 |
| | | | QuaRot | $0.0_{\pm0.0}$ | $1.0_{\pm0.6}$ | $0.7_{\pm0.1}$ | $22.9_{\pm5.3}$ | $0.0_{\pm0.0}$ | $4.9_{\pm1.2}$ | -58.9 |
| | | 16-16-3 | KVQuant* | $33.6_{\pm3.9}$ | $91.5_{\pm0.8}$ | $90.5_{\pm0.6}$ | $46.1_{\pm0.8}$ | $28.1_{\pm3.1}$ | $58.0_{\pm1.4}$ | -5.9 |
| | | | QuaRot | $0.3_{\pm0.5}$ | $0.8_{\pm0.2}$ | $0.4_{\pm0.3}$ | $24.6_{\pm1.8}$ | $0.0_{\pm0.0}$ | $5.2_{\pm0.4}$ | -58.6 |
| | *Weight-Act.* | 8-8-8 | SmoothQuant | $45.0_{\pm3.8}$ | $94.0_{\pm0.7}$ | $90.8_{\pm0.3}$ | $50.7_{\pm1.3}$ | $35.7_{\pm1.1}$ | $63.2_{\pm0.6}$ | -0.6 |
| | | | FlatQuant | $47.8_{\pm2.7}$ | $93.8_{\pm0.8}$ | $91.0_{\pm0.6}$ | $49.3_{\pm0.6}$ | $36.2_{\pm1.0}$ | $63.6_{\pm0.5}$ | -0.2 |
| | | 4-4-4 | MXFP4 | $0.0_{\pm0.0}$ | $0.5_{\pm0.1}$ | $0.2_{\pm0.1}$ | $20.0_{\pm2.2}$ | $0.0_{\pm0.0}$ | $4.1_{\pm0.5}$ | -59.7 |
| | | | FlatQuant | $26.1_{\pm1.3}$ | $84.1_{\pm1.3}$ | $90.9_{\pm0.4}$ | $46.6_{\pm4.6}$ | $12.4_{\pm2.3}$ | $52.0_{\pm0.9}$ | -11.8 |
| DeepSeek-R1-Distill-Qwen-14B | **BF16** | - | - | $54.7_{\pm0.5}$ | $95.5_{\pm0.4}$ | $93.7_{\pm0.2}$ | $62.6_{\pm1.8}$ | $50.9_{\pm0.8}$ | $71.5_{\pm0.2}$ | - |
| | *Weight-only* | 4-16-16 | AWQ | $53.9_{\pm1.7}$ | $94.7_{\pm0.8}$ | $93.3_{\pm0.6}$ | $60.6_{\pm1.0}$ | $49.6_{\pm0.8}$ | $70.4_{\pm0.2}$ | -1.1 |
| | | | GPTQ | $52.5_{\pm2.2}$ | $94.9_{\pm0.3}$ | $93.7_{\pm0.3}$ | $60.3_{\pm2.0}$ | $48.3_{\pm0.6}$ | $69.9_{\pm0.3}$ | -1.6 |
| | | 3-16-16 | AWQ | $42.2_{\pm1.3}$ | $94.1_{\pm0.6}$ | $93.0_{\pm0.4}$ | $53.7_{\pm2.3}$ | $43.3_{\pm1.0}$ | $65.3_{\pm0.7}$ | -6.2 |
| | | | GPTQ | $47.2_{\pm4.7}$ | $94.1_{\pm0.2}$ | $93.5_{\pm0.4}$ | $55.2_{\pm1.5}$ | $44.8_{\pm2.0}$ | $67.0_{\pm1.5}$ | -4.5 |
| | *KV Cache* | 16-16-4 | KVQuant* | $55.3_{\pm4.2}$ | $94.7_{\pm0.6}$ | $93.8_{\pm0.2}$ | $61.3_{\pm1.9}$ | $49.6_{\pm0.4}$ | $70.9_{\pm0.8}$ | -0.6 |
| | | | QuaRot | $54.7_{\pm3.2}$ | $95.0_{\pm0.5}$ | $93.8_{\pm0.3}$ | $60.9_{\pm1.9}$ | $52.6_{\pm1.5}$ | $71.4_{\pm0.8}$ | -0.1 |
| | | 16-16-3 | KVQuant* | $42.8_{\pm3.2}$ | $93.2_{\pm0.7}$ | $93.1_{\pm0.3}$ | $56.1_{\pm1.3}$ | $45.5_{\pm1.3}$ | $66.1_{\pm0.7}$ | -5.4 |
| | | | QuaRot | $52.8_{\pm3.2}$ | $95.1_{\pm0.5}$ | $93.5_{\pm0.1}$ | $60.1_{\pm1.8}$ | $44.2_{\pm2.7}$ | $69.1_{\pm0.9}$ | -2.4 |
| | *Weight-Act.* | 8-8-8 | SmoothQuant | $59.4_{\pm1.3}$ | $95.5_{\pm0.4}$ | $94.0_{\pm0.4}$ | $60.9_{\pm2.4}$ | $50.6_{\pm1.8}$ | $72.1_{\pm0.5}$ | 0.6 |
| | | | QuaRot | $56.7_{\pm2.5}$ | $95.2_{\pm0.7}$ | $94.0_{\pm0.3}$ | $58.8_{\pm3.0}$ | $51.0_{\pm0.6}$ | $71.1_{\pm0.2}$ | -0.4 |
| | | | FlatQuant | $57.5_{\pm0.8}$ | $95.1_{\pm0.1}$ | $93.8_{\pm0.2}$ | $61.3_{\pm3.8}$ | $50.8_{\pm1.0}$ | $71.7_{\pm0.7}$ | 0.2 |
| | | 4-4-4 | MXFP4 | $23.3_{\pm4.2}$ | $87.5_{\pm0.8}$ | $91.3_{\pm0.4}$ | $47.5_{\pm3.6}$ | $32.8_{\pm2.0}$ | $56.5_{\pm1.1}$ | -15.0 |
| | | | QuaRot | $42.5_{\pm4.4}$ | $92.7_{\pm0.9}$ | $93.0_{\pm0.3}$ | $55.7_{\pm3.0}$ | $45.7_{\pm1.3}$ | $65.9_{\pm0.5}$ | -5.6 |
| | | | FlatQuant | $50.3_{\pm2.4}$ | $94.7_{\pm0.3}$ | $93.3_{\pm0.1}$ | $55.7_{\pm2.0}$ | $48.4_{\pm1.2}$ | $68.5_{\pm0.4}$ | -3.0 |
| DeepSeek-R1-Distill-Qwen-32B | **BF16** | - | - | $61.7_{\pm1.7}$ | $96.3_{\pm0.5}$ | $94.2_{\pm0.2}$ | $65.7_{\pm1.8}$ | $56.0_{\pm2.3}$ | $74.8_{\pm0.4}$ | - |
| | *Weight-only* | 4-16-16 | AWQ | $63.6_{\pm1.3}$ | $95.9_{\pm0.6}$ | $94.4_{\pm0.2}$ | $63.5_{\pm1.9}$ | $54.8_{\pm1.5}$ | $74.4_{\pm0.6}$ | -0.4 |
| | | | GPTQ | $57.5_{\pm1.4}$ | $95.9_{\pm0.8}$ | $94.3_{\pm0.1}$ | $62.3_{\pm1.5}$ | $54.2_{\pm0.4}$ | $72.9_{\pm0.5}$ | -1.9 |
| | | 3-16-16 | AWQ | $53.1_{\pm1.7}$ | $94.1_{\pm1.1}$ | $94.1_{\pm0.1}$ | $62.6_{\pm1.5}$ | $50.4_{\pm1.6}$ | $70.9_{\pm1.0}$ | -3.9 |
| | | | GPTQ | $51.7_{\pm4.4}$ | $94.7_{\pm0.3}$ | $94.1_{\pm0.2}$ | $57.6_{\pm2.7}$ | $50.1_{\pm1.5}$ | $69.6_{\pm1.0}$ | -5.1 |
| | *KV Cache* | 16-16-4 | KVQuant* | $57.5_{\pm1.4}$ | $96.0_{\pm0.4}$ | $94.7_{\pm0.4}$ | $64.3_{\pm1.9}$ | $55.9_{\pm0.8}$ | $73.7_{\pm0.5}$ | -1.1 |
| | | | QuaRot | $63.1_{\pm1.9}$ | $95.8_{\pm0.2}$ | $94.2_{\pm0.4}$ | $63.1_{\pm1.8}$ | $56.3_{\pm1.1}$ | $74.5_{\pm0.9}$ | -0.3 |
| | | 16-16-3 | KVQuant* | $50.8_{\pm4.2}$ | $94.8_{\pm0.9}$ | $94.4_{\pm0.1}$ | $62.3_{\pm2.1}$ | $51.6_{\pm0.6}$ | $70.8_{\pm0.8}$ | -4.0 |
| | | | QuaRot | $57.2_{\pm2.7}$ | $94.9_{\pm0.5}$ | $94.1_{\pm0.5}$ | $63.8_{\pm1.1}$ | $48.6_{\pm2.3}$ | $71.7_{\pm0.8}$ | -3.0 |
| | *Weight-Act.* | 8-8-8 | SmoothQuant | $59.2_{\pm2.2}$ | $95.4_{\pm0.4}$ | $94.2_{\pm0.3}$ | $64.0_{\pm1.2}$ | $56.7_{\pm1.7}$ | $73.9_{\pm0.5}$ | -0.9 |
| | | | QuaRot | $56.9_{\pm2.6}$ | $95.3_{\pm0.3}$ | $94.3_{\pm0.3}$ | $63.0_{\pm0.8}$ | $55.5_{\pm1.7}$ | $73.0_{\pm0.7}$ | -1.7 |
| | | | FlatQuant | $62.2_{\pm3.4}$ | $95.9_{\pm0.6}$ | $94.4_{\pm0.3}$ | $65.2_{\pm0.5}$ | $56.1_{\pm0.6}$ | $74.7_{\pm0.8}$ | -0.0 |
| | | 4-4-4 | MXFP4 | $35.3_{\pm3.5}$ | $92.4_{\pm0.4}$ | $93.7_{\pm0.6}$ | $57.1_{\pm3.6}$ | $36.2_{\pm0.7}$ | $62.9_{\pm0.7}$ | -11.8 |
| | | | QuaRot | $46.9_{\pm2.1}$ | $94.1_{\pm0.4}$ | $93.9_{\pm0.2}$ | $60.1_{\pm2.2}$ | $49.4_{\pm0.6}$ | $68.9_{\pm0.3}$ | -5.9 |
| | | | FlatQuant | $57.8_{\pm4.3}$ | $95.1_{\pm0.6}$ | $94.2_{\pm0.0}$ | $58.1_{\pm2.2}$ | $54.2_{\pm1.3}$ | $71.9_{\pm1.6}$ | -2.9 |

Table 1: The overall assessment of quantized DeepSeek-R1-Distill-Qwen models on various reasoning benchmarks. QuaRot for the 7B model weight-activation quantization is skipped because of the incompatible hidden dimension for Hadamard transformation. The green, orange and red cells stand for the lossless ($\leq1\%$), the fair ($1\%$-$3\%$) and the risky ($\geq3\%$) respectively. Note that 1.5B and 7B models are exceptions with severe degradation for low-bit KV cache and weight-activation quantization, as will be discussed in Appendix C.2. Results on more models are available in Appendix B.

**For weight-only quantization, 4-bit is nearly lossless, but 3-bit becomes risky.** From Table 1, quantizing the weights of the distilled Qwen models to 4 bits incurs a slight performance drop compared to their BF16 counterparts, i.e., 2.1% and 0.4% for the 1.5B and 32B models, respectively. However, when reducing to 3 bits, there is a sharp drop in performance for each model, e.g., over 7% accuracy drop for the 1.5B model and 3% for the 32B model. Meanwhile, smaller reasoning models tend to suffer more from quantization, which aligns past experiences on non-reasoning models (Li et al., 2024). Similar observations can be found on other sizes and families of LLMs in Appendix B.1.

**For KV cache, we suggest 4-bit quantization.** We find that 4-bit quantized KV cache achieves lossless performance on large models (e.g., 14B or 32B). In particular, the accuracy drop can remain smaller than 1% even on AIME-120, where the model generates lengthy output of over 10K tokens, as will be discussed in §4.1. In addition, the performance of smaller LLMs is also relatively fair, with an average drop of 1.4% on 1.5B and 1.0% on 7B, respectively. However, for more aggressive KV cache quantization, the performance incurs a sharp drop, especially for smaller LLMs, i.e., smaller models (1.5B and 7B) and larger models (14B and 32B) incur over 5% and 2% accuracy drop when the bitwidth of KV cache quantization reduces to 3 bits, respectively.

**We recommend 8-bit quantization for weights, activation, and KV cache, which is lossless across various LLM sizes and reasoning tasks.** It can be observed that for all reasoning models and tasks evaluated, the best-performing W8A8KV8 quantization algorithm achieves a performance drop of less than 1 point. This holds true even for the smallest DeepSeek-R1-Distill-Qwen-1.5B model. Nevertheless, when we apply more aggressive quantization with 4 bits, even the large 32B model incurs an accuracy drop of 2.9%. The degradation becomes significantly more pronounced for smaller LLMs, with accuracy drops exceeding 10% on the 1.5B and 7B models. Thus, improving 4-bit weight-activation quantization for reasoning models remains an open challenge for the community.

### 3.3 Comparisons of Quantization Algorithms

**AWQ is preferred over GPTQ for weight quantization, considering its comparable performance but more efficient implementation.** According to Table 1, AWQ performs on par with GPTQ for both 4-bit and 3-bit quantization. For instance, it surpassed GPTQ three times and lost once for 4-bit quantization over the evaluated models. In addition, AWQ is usually faster to implement compared with GPTQ, since it does not require the iterative update of LLM parameters. Besides, AWQ is also more robust to the choice of calibration data than GPTQ, as discussed in Section 4.2. We thus recommend AWQ as the default algorithm for the weight-only quantization of reasoning models.

**QuaRot is generally preferred over KVQuant* for KV cache Quantization, except for Qwen 1.5B and 7B models which have unexpected huge outliers over bias.** According to Table 1, QuaRot generally outperforms KVQuant* on the 14B and 32B models. For instance, on the DeepSeek-R1-Distill-Qwen-32B model, QuaRot with 3-bit quantization achieves an average performance of 71.7%, while KVQuant* only reaches 70.8%. However, exceptions are observed in the 1.5B and 7B models, where KVQuant* significantly surpasses QuaRot. We find that this is due to the huge outliers over biases in the key and value layers of these two models. While KVQuant* can mitigate this by quantizing the output before bias, it is not applicable for QuaRot. More discussions can be found in Appendix C.2.

**For 8-bit weight-activation quantization, SmoothQuant proves sufficient, while FlatQuant emerges as the leading algorithm for 4-bit quantization.** In the 8-bit scenario, all algorithms achieve near-lossless results across most models, with no clear leading algorithm. We therefore recommend SmoothQuant for 8-bit quantization due to its zero-overhead advantage. However, in the more challenging 4-bit setting, FlatQuant demonstrates significant superiority over competing algorithms while introducing only minimal latency overhead. For instance, FlatQuant achieves an average performance of 71.9% on the 32B model, with a performance drop of only 2.9%, significantly surpassing both MXFP4 (11.8%↓) and QuaRot (5.9%↓). Note that substantial accuracy loss still exists under W4A4KV4 quantization on 1.5B and 7B models even for FlatQuant. We analyze its reason in Appendix C.2.

In the rest of this study, we adopt AWQ for weight-only quantization, QuaRot for KV cache quantization, and FlatQuant for weight-activation quantization if not otherwise specified.

### 3.4 Impacts of Task Difficulty

**Quantization is prone to fail on more difficult tasks.** Based on Table 1, we compare the performance drop caused by quantization on the three math reasoning benchmarks with varying difficulty levels, and find that more difficult tasks incur a more severe performance

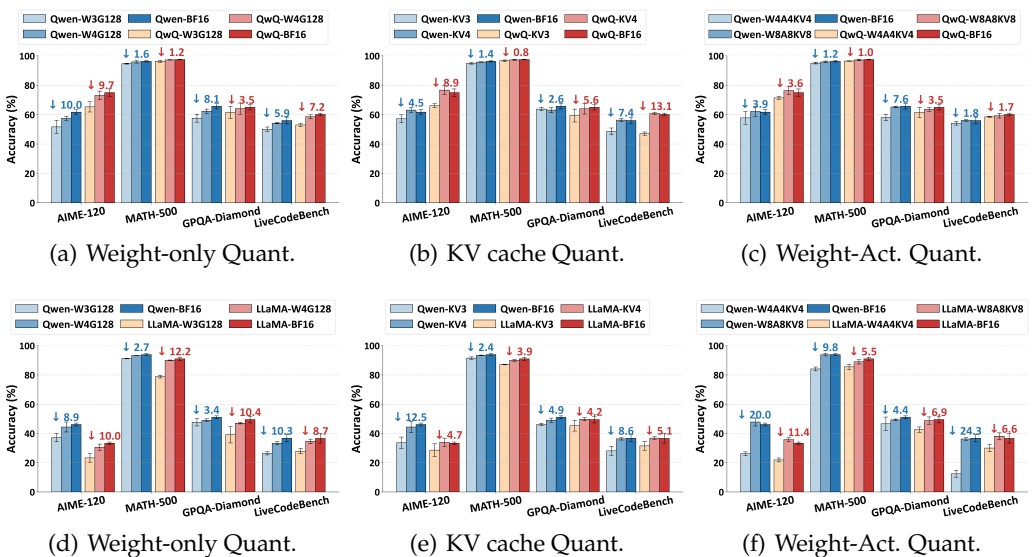

Figure 2: (a)-(c) shows the comparisons between DeepSeek-R1-Distill-Qwen-32B (SFT-based) and QwQ-32B (RL-based) on different quantization configurations. (d)-(f) shows comparisons between DeepSeek-R1-Distill-Qwen-7B and DeepSeek-R1-Distill-LLaMA-8B on different quantization configurations.

drop. Among them, AIME-120 is the most challenging task, with samples selected from American Math Competition questions. MATH-500 contains a mix of easy and hard problems, followed by GSM8K with easy primary school-level questions. The performance degradations are in descending order among the three benchmarks. For instance, for the 32B model with W4A4KV4 quantization, the performance drops are 3.9% on AIME-120, 1.2% on MATH-500, and 0.0% on GSM8K.

In addition to Table 1, we also examine the accuracy at the 5 different difficulty levels explicitly defined by MATH-500 (Lightman et al., 2023) in Appendix B.2, where the observations remain consistent.

## 3.5 The Impacts of the Origin of Reasoning LLMs

In this section, we compare the quantization performance on reasoning models trained from different methods, with the suggested algorithms in § 3.3. Specifically, we study

1. the reasoning models obtained with either distillation (i.e., DeepSeek-R1-Distill-Qwen-32B) and reinforcement learning (i.e., QwQ-32B); and
2. different LLM families (i.e., LLaMA and Qwen) distilled from DeepSeek-R1.

**Qwen-32B is more robust to KV cache quantization, while QwQ-32B is more resilient to weight-only and weight-activation quantization.** As seen in Figure 2 (a)-(c), when comparing Qwen-32B and QwQ-32B, we observe that the models exhibit varying tolerance to different quantization configurations. For instance, Qwen-32B experiences a larger accuracy drop of 6.4% for W3G128 and 3.8% for KV3 compared to QwQ-32B's 5.4% and 2.5%, respectively. However, Qwen-32B gains an edge in KV cache quantization, with 3.1% less quantization loss. We hypothesize that although both models originate from the same base model, differences in their training recipes lead to distinct training dynamics, shaping their intermediate representations differently and ultimately influencing quantization accuracy.

**Qwen-7B is more robust to weight-only quantization, while LLaMA-8B is more resilient to KV cache and weight-activation quantization.** To study the impact of LLM origins, we compared DeepSeek-R1-Distill-Qwen-7B and DeepSeek-R1-Distill-LLaMA-8B, which

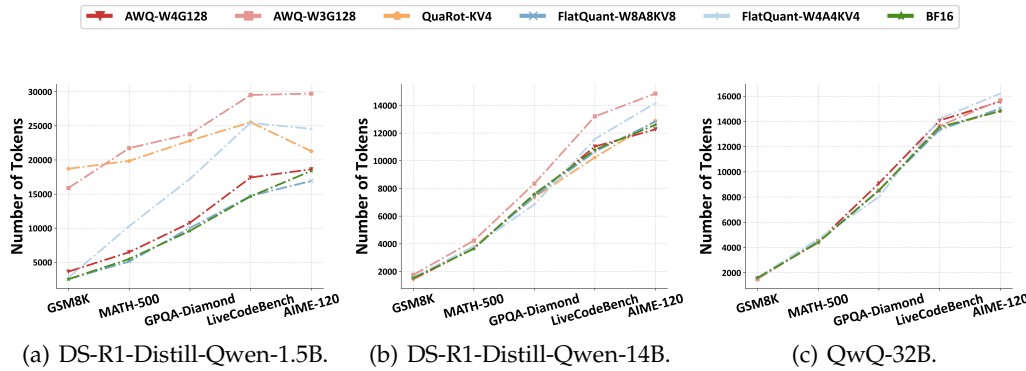

(a) DS-R1-Distill-Qwen-1.5B.    (b) DS-R1-Distill-Qwen-14B.    (c) QwQ-32B.

Figure 3: The lengths of generated tokens across different datasets and models.

are of similar size and distilled from DeepSeek-R1 with the same data. As shown in Figure 2 (d)-(f), Qwen-7B is more robust to weight-only quantization (e.g., 8.9% vs. 10.0% for W3G128 on AIME-120). However, it is more sensitive for KV cache and activation quantization (e.g., 12.5% vs. 4.7% drop for KV3 on AIME-120, and 20.0% vs. 11.4% drop for W4A4KV4 on AIME-120). Potential reasons include differences in pretraining data, fine-tuning configurations, and training dynamics.

More detailed results of QwQ-32B and DeepSeek-R1-Distill-LLaMA for different quantization configurations and algorithms can be found in Appendix B.1.

## 4 Discussions

In this section, we provide further analysis for quantized reasoning models, including their output length, choice of calibration data, scaling effects w.r.t both model size and latency, as well as test-time scaling. The analysis is based on the best-performing quantization algorithms according to Section 3.3. Due to limited space, we provide more analysis on the choice of calibration data in Appendix C.1, quantization of DeepSeek-R1-Distill-Qwen-1.5B and 7B model in Appendix C.2, and qualitative examples in Appendix D.

### 4.1 Do Quantized Reasoning Models Think Longer?

Since quantization inherently reduces model capacity due to precision loss, one concern with quantized LLMs is whether quantized models need to think longer to compensate for the reduced representation power, which could prolong the end-to-end latency. We analyze the output length statistics of quantized reasoning models in Figure 3. **It is found that for various quantization configurations (e.g., W4G128, W8A8KV8, and KV4), the generated output lengths closely align with those of the BF16 models.** However, under more aggressive low-bit quantization (e.g., W3G128 and W4A4KV4), LLMs tend to produce longer outputs. This effect is particularly pronounced in smaller models like DeepSeek-R1-Distill-Qwen-1.5B, where lower-bit quantization results in more pronounced performance degradation, as previously discussed in Table 1.

While more difficult tasks tend to generate longer outputs, they do not inherently trigger longer reasoning steps in quantized LLMs. Similar observations are provided in Appendix B.3, where we analyze the output lengths across five difficulty levels on MATH-500.

### 4.2 Choice of Calibration Data

Calibration data has always been an important part of post-training quantization. While previous studies (Williams & Aletras, 2023; Lin et al., 2023) suggest that the overall performance of quantization methods on pre-trained and instruction-tuned models is robust to the source domain of calibration data, we find this is not true for reasoning models. Instead,

| Methods | W-A-KV # Bits | Calibration Domains | AIME-120 | MATH-500 | GSM8K | GPQA-Diamond | LiveCode-Bench | Avg. | Δ |
|---|---|---|---|---|---|---|---|---|---|
| BF16 | - | - | 21.7 | 84.4 | 84.6 | 36.9 | 16.0 | 48.7 | - |
| AWQ | 3-16-16 | WikiText2 | 5.8 | 54.4 | 68.4 | 31.8 | 3.7 | 32.8 | 1.8 |
|  |  | Numina-Math-1.5 | 5.8 | 60.2 | 71.3 | 28.8 | 6.7 | 34.6 |  |
| GPTQ | 3-16-16 | WikiText2 | 3.3 | 49.4 | 57.5 | 27.3 | 3.7 | 28.3 | 9.8 |
|  |  | Numina-Math-1.5 | 10.0 | 71.6 | 75.7 | 23.7 | 9.3 | 38.1 |  |
| KVQuant* | 16-16-4 | WikiText2 | 20.8 | 84.6 | 84.0 | 33.8 | 13.4 | 47.3 | 0.4 |
|  |  | Numina-Math-1.5 | 20.0 | 83.8 | 84.5 | 33.8 | 16.4 | 47.7 |  |
| SmoothQuant | 8-8-8 | WikiText2 | 17.5 | 79.8 | 83.2 | 34.3 | 15.7 | 46.1 | 0.9 |
|  |  | Numina-Math-1.5 | 20.8 | 79.8 | 82.7 | 35.9 | 15.7 | 47.0 |  |
| FlatQuant | 4-4-4 | WikiText2 | 10.0 | 64.8 | 78.6 | 31.8 | 6.7 | 38.4 | 0.2 |
|  |  | Numina-Math-1.5 | 8.3 | 65.6 | 78.0 | 33.3 | 7.5 | 38.6 |  |

Table 2: The impact of calibration data domain on different quantization methods. The evaluation is conducted on the quantized DeepSeek-R1-Distill-Qwen-1.5B model across various reasoning benchmarks. The green, orange and red cells stand for the robust ($\leq$1%), the fair (1%-3%) and the sensitive ($\geq$3%) quantization methods respectively.

the choice of calibration data domain may have a huge impact on the quantization accuracy. In addition, this effect is highly dependent on the quantization method used. Due to limited space, we leave more details in Appendix C.1, including the analysis on the distributions of calibration data from different domains.

**The domain of calibration data affects GPTQ, but not the rest of the quantization algorithms.** Prior works (Frantar et al., 2022; Lin et al., 2023) on post-training quantization usually sample calibration data from pre-training datasets like WikiText2 (Merity et al., 2016) or C4 (Raffel et al., 2020). However, we find that using reasoning data for calibration is crucial for quantization methods that rely heavily on the calibration data for quantization error compensation (e.g. GPTQ (Frantar et al., 2022)). In Figure 9, we visualize the activation distributions from different source domains. It can be seen that domain gaps exist between reasoning (i.e. Numina-Math-1.5 and LiveCodeBench) and pre-training (i.e. WikiText2) data, which can cause training-inference inconsistency issues that seriously deteriorate the quantized model if it is calibrated on the pre-training data. As shown in Table 2, switching the GPTQ calibration set from WikiText2 to Numina-Math-1.5 leads to an average accuracy gain of 9.8%. Also, note that the source domain of the calibration data has a relatively small impact on other quantization methods such as KVQuant* (Hooper et al., 2024), and FlatQuant (Sun et al., 2024). These methods only rely on calibration data for quantization parameter computation or outlier channel pattern identification. Given the high consistency of channel-wise magnitude distribution across different domains as shown in Figure 8, these methods are more robust to domain shift than GPTQ.

## 4.3 Scaling Effects of Quantized Reasoning Models

In this section, we study the performance change when we scale the size and latency of the quantized model. Specifically, we aim to identify cost-effective quantization configurations that optimally balance accuracy with model size or latency.

**Model Size Scaling.** In Figure 4(a), we present the trade-off between accuracy and model size based on the series of DeepSeek-R1-Distill Qwen and QwQ-32B. We evaluate different weight-only quantization bit-widths with AWQ (i.e., 8-bit, 4-bit, and 3-bit) against the BF16 model, and report the accuracy on LiveCodeBench. It can be found that as the model size increases, accuracy improves rapidly. Under the same model size, the large LLMs in low precisions are more accurate than small LLMs in BF16. For instance, while both the 3-bit quantized QwQ-32B and Qwen-32B are similar in size (i.e., ~14.5GB) with Qwen-7B in BF16, they significantly outperform Qwen-7B by around 17%.

**Latency Scaling.** While large models demonstrate better accuracy-size trade-off, they are inevitably slower for inference. Here we also study the scaling effect of latency w.r.t. the accuracy. The latency is measured as the end-to-end time cost over the LiveCodeBench

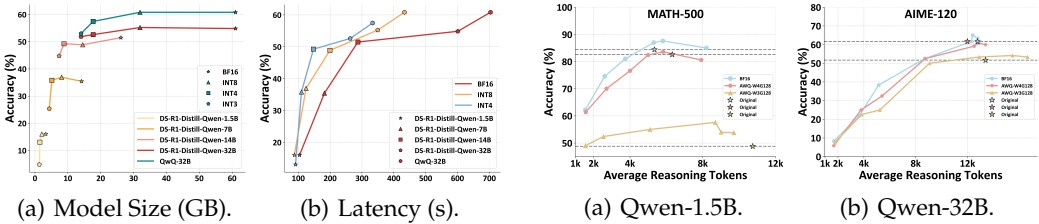

(a) Model Size (GB).     (b) Latency (s).     (a) Qwen-1.5B.     (b) Qwen-32B.

Figure 4: Model size and average latency *v.s.* accuracy (%) on LiveCodeBench.

Figure 5: Average reasoning tokens *v.s.* accuracy (%) on DS-R1-Distill-Qwen models.

dataset, i.e., the multiplication between the number of generated tokens with the Time Per Output Token (TPOT)[1]. According to Figure 4(b), 4-bit quantized models provide superior accuracy–latency trade-off than the 8-bit and BF16 counterparts, which mainly benefits from increased generation speed while maintaining the same reasoning length.

## 4.4 Test-time Scaling of Quantized Reasoning Models

Test-time scaling has become a prevalent technique for enhancing reasoning model performance (Snell et al., 2024; Muennighoff et al., 2025; Liu et al., 2025). To investigate the behavior of quantized reasoning models under test-time scaling, we adopt the method of Muennighoff et al. (2025) by controlling reasoning length through dynamic adjustment of average reasoning tokens. For suppression, early termination forces partial reasoning integration; for promotion, "wait" tokens extend reasoning until token budgets are met.

From Figure 5, longer reasoning lengths can generally enhance the performance of each model. For example, on MATH-500, both AWQ-W4G128 and AWQ-W3G128 exhibit clear improvements with extended reasoning. However, the increasing rates of quantized LLMs are generally lower than that of BF16. We speculate that it is due to the quantization error accumulated along the sequence, which hinders effective reasoning. In addition, excessively long reasoning lengths also result in performance degradation, as also discovered in Muennighoff et al. (2025). The output of unnecessary steps could degrade overall accuracy. These findings highlight the importance of only scaling the reasoning length within an appropriate range. How to achieve consistent improvement over a longer range is an open challenge.

## 5 Conclusion

In this study, we present the first systematic study on the quantization of reasoning models. We examine the performance of different state-of-the-art quantization methods on different reasoning models under various quantization configurations. We also recommend the lossless quantization settings like W8A8 and W4A16 quantization, as well as the corresponding preferred quantization methods. Additionally, we find that task difficulty and model origins have a large impact on the quantized model's performance. In particular, more difficult tasks incur more severe performance degradation caused by quantization. Besides, both the training recipes and model families influence the quantization sensitivity. Like the BF16 models, quantized models also show consistent improvement when gradually increasing test-time compute budget over a certain range, but may behave worse afterward. While our study identifies critical factors influencing the performance of quantized reasoning models, the underlying mechanisms driving these effects remain incompletely understood. Future work will focus on elucidating these causal relationships while advancing systematic methods to optimize the accuracy-efficiency trade-off in reasoning systems.

---

[1]For TPOT, we refer to the official latency report by Qwen: `https://qwen.readthedocs.io/en/latest/benchmark/speed_benchmark.html`

## 6 Acknowledgment

We sincerely thank Dan Alistarh and Eldar Kurtić for their helpful suggestions in improving our work. We also appreciate the valuable feedback from all anonymous reviewers during the peer-review process.

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

# A  Quantization Algorithms and Implementation Details

Below, we briefly review the quantization algorithms implemented in this study, including weight-only quantizaiton, KV-Cache quantization, and weight-activation quantization.

## A.1  Weight-only Quantization

**GPTQ** (Frantar et al., 2022) aims to construct the optimal weights $\hat{\mathbf{W}}^*$ that minimize the squared error between full-precision and quantized outputs given input $\mathbf{X}$, i.e., $\arg\min_{\hat{\mathbf{W}}} \|\hat{\mathbf{W}}\mathbf{X} - \mathbf{W}\mathbf{X}\|_F$. Specifically, it iteratively quantizes weights and updates the remaining ones using approximate second-order information derived from the inverse Hessian matrix. This process is optimized via Cholesky decomposition to enhance numerical stability and computational efficiency. In this work, we adopt asymmetric quantization with a group size of 128, i.e., the step size is calculated per 128 consecutive elements per channel in $\mathbf{W}$. We use activation reorder to deal with outlier channels, with static group activated to reduce latency overhead except for DeepSeek-R1-Distill-LLaMA-70B model. To mitigate the effects of domain shift as detailed in Appendix C.1, we use reasoning models to self-generate reasoning data on Numina-Math-1.5 (LI et al., 2024) dataset, and randomly sample 128 text sequences of length 2048 to construct the calibration set.

**AWQ** (Lin et al., 2023) aims to identify and protect salient weights to improve the quantized LLMs. Since activations $\mathbf{X}$ of LLMs are rich in outliers, AWQ employs per-channel scaling, e.g., $\mathbf{Y} = (\mathbf{X} \cdot \boldsymbol{c}^{-1})(\boldsymbol{c} \cdot \mathbf{W}^\top)$, where the channel-wise scaling factor $\boldsymbol{c} \in \mathbb{R}^n$ can be analytically derived to balance the magnitudes of input activations and weights. The optimal $\boldsymbol{c}$ can be obtained by $\boldsymbol{c} = \boldsymbol{c}_\mathbf{X}^\alpha \cdot \boldsymbol{c}_\mathbf{W}^{-\beta}$, where $c_\mathbf{X}$ and $c_\mathbf{W}$ are the average channel statistics of activations and weights, and $\alpha, \beta \in [0,1]$ are the associated hyper-parameters. Moreover, the scaled weights $\boldsymbol{c} \cdot \mathbf{W}^\top$ can be merged together to eliminate runtime overhead. Similar to GPTQ, we adopt asymmetric quantization with a group size of 128. For the calibration data, we follow Lin et al. (2023) and sample 128 text sequences of length 512 from Pile (Gao et al., 2020) dataset.

## A.2  KV-Cache Quantization

**KVQuant** (Hooper et al., 2024) leverages the outlier channels in K cache, proposing to quantize K Cache with static per-channel quantization. To better preserve the outlier patterns of K cache, it quantizes K cache before applying RoPE (Su et al., 2024). Additionally, to accommodate the non-uniform distribution of the KV cache, it incorporates non-uniform and mixed-precision quantization. In this work, we skip the non-uniform and mixed-precision quantization in KVQuant and implement KVQuant* as a simplified baseline, employing static per-channel quantization for K cache and dynamic per-token quantization for V cache with a group size of 128. For DeepSeek-R1-Distill-Qwen-1.5B and 7B models, we further incorporate pre-bias K cache quantization to suppress the extreme outlier channels in K cache, with more details in Appendix C.2. We randomly sample 128 text sequences of length 512 from Pile (Gao et al., 2020) dataset as the calibration set.

**QuaRot** (Ashkboos et al., 2024) can be applied for both KV cache quantization and weight-activation quantization. It enables end-to-end 4-bit inference by using Hadamard transformations to eliminate outliers in activations and KV caches. Based on the orthogonality of Hadamard matrices (i.e., $\mathbf{H}^\top\mathbf{H} = \mathbf{I}$), the output of quantized linear layer is $\hat{\mathbf{Y}} = \mathcal{Q}(\mathbf{X}\mathbf{H}) \cdot \mathcal{Q}(\mathbf{H}^\top\mathbf{W}^\top)$, where the transformed weight $\mathbf{W}\mathbf{H}$ can be pre-processed offline to reduce additional runtime overhead. It also integrates GPTQ (Frantar et al., 2022) to reduce quantization error on weights. For KV cache quantization, we use per-token asymmetric quantization with a group size of 128. For weight-activation quantization, we apply per-channel weight symmetric quantization and per-token activation asymmetric quantization for computing efficiency. We leave the queries in full precision to leverage the memory-bound nature of self-attention operation following Ashkboos et al. (2024).

| Model | Quantization | W-A-KV # Bits | Methods | AIME-120 | MATH-500 | GSM8K | GPQA-Diamond | LiveCode-Bench | Avg. | Drop↓ |
|---|---|---|---|---|---|---|---|---|---|---|
| QwQ-32B | BF16 | - | - | $75.0_{\pm2.5}$ | $97.5_{\pm0.2}$ | $95.6_{\pm0.2}$ | $65.0_{\pm1.8}$ | $60.2_{\pm0.8}$ | $78.7_{\pm0.8}$ | - |
| | Weight-only | 4-16-16 | AWQ | $73.3_{\pm2.9}$ | $97.3_{\pm0.5}$ | $95.7_{\pm0.3}$ | $64.3_{\pm1.8}$ | $58.7_{\pm0.6}$ | $77.9_{\pm1.1}$ | -0.8 |
| | | | GPTQ | $73.1_{\pm2.9}$ | $97.3_{\pm0.2}$ | $95.5_{\pm0.2}$ | $64.1_{\pm3.8}$ | $58.6_{\pm1.4}$ | $77.7_{\pm0.3}$ | -0.9 |
| | | 3-16-16 | AWQ | $60.0_{\pm1.4}$ | $96.1_{\pm0.5}$ | $95.4_{\pm0.2}$ | $57.1_{\pm1.3}$ | $50.3_{\pm1.1}$ | $71.8_{\pm0.6}$ | -6.9 |
| | | | GPTQ | $65.3_{\pm3.5}$ | $96.3_{\pm0.6}$ | $95.5_{\pm0.0}$ | $61.5_{\pm4.1}$ | $53.0_{\pm1.0}$ | $74.3_{\pm1.0}$ | -4.4 |
| | KV Cache | 16-16-4 | KVQuant* | $75.8_{\pm0.0}$ | $97.7_{\pm0.2}$ | $95.3_{\pm0.2}$ | $65.0_{\pm0.6}$ | $58.0_{\pm1.1}$ | $78.4_{\pm0.1}$ | -0.3 |
| | | | QuaRot | $76.7_{\pm2.9}$ | $97.3_{\pm0.3}$ | $95.7_{\pm0.2}$ | $64.0_{\pm3.6}$ | $60.8_{\pm0.7}$ | $78.9_{\pm0.8}$ | 0.3 |
| | | 16-16-3 | KVQuant* | $60.3_{\pm1.0}$ | $95.7_{\pm0.1}$ | $95.5_{\pm0.6}$ | $58.8_{\pm1.2}$ | $54.7_{\pm0.6}$ | $73.0_{\pm0.6}$ | -5.7 |
| | | | QuaRot | $66.1_{\pm1.3}$ | $96.7_{\pm0.5}$ | $95.0_{\pm0.5}$ | $59.4_{\pm4.4}$ | $47.1_{\pm1.1}$ | $72.9_{\pm0.3}$ | -5.8 |
| | Weight-Act. | 8-8-8 | SmoothQuant | $75.8_{\pm0.8}$ | $98.0_{\pm0.4}$ | $95.7_{\pm0.0}$ | $65.5_{\pm5.1}$ | $60.0_{\pm1.2}$ | $79.0_{\pm1.2}$ | 0.3 |
| | | | QuaRot | $73.6_{\pm1.0}$ | $97.2_{\pm0.6}$ | $95.7_{\pm0.2}$ | $65.7_{\pm2.3}$ | $60.8_{\pm1.0}$ | $78.6_{\pm0.5}$ | -0.1 |
| | | | FlatQuant | $76.4_{\pm2.7}$ | $97.2_{\pm0.5}$ | $95.8_{\pm0.4}$ | $63.6_{\pm1.3}$ | $59.3_{\pm1.6}$ | $78.5_{\pm0.7}$ | -0.2 |
| | | 4-4-4 | MXFP4 | $34.4_{\pm4.1}$ | $92.7_{\pm0.4}$ | $94.6_{\pm0.2}$ | $55.1_{\pm2.7}$ | $34.5_{\pm1.7}$ | $62.2_{\pm0.5}$ | -16.4 |
| | | | QuaRot | $55.3_{\pm4.1}$ | $96.0_{\pm0.4}$ | $94.9_{\pm0.0}$ | $58.4_{\pm1.6}$ | $51.2_{\pm1.8}$ | $71.2_{\pm1.3}$ | -7.5 |
| | | | FlatQuant | $71.4_{\pm1.0}$ | $96.5_{\pm0.2}$ | $95.5_{\pm0.2}$ | $61.5_{\pm3.4}$ | $58.5_{\pm0.4}$ | $76.7_{\pm0.5}$ | -2.0 |

Table 3: The overall assessment of quantized QwQ-32B model on various reasoning benchmarks. The green, orange and red cells stand for the lossless ($\leq$1%), the fair (1%-3%) and the risky ($\geq$3%) respectively.

## A.3 Weight-Activation Quantization

**SmoothQuant** (Xiao et al., 2023) employs per-channel scaling, i.e., $\mathbf{Y} = (\mathbf{X} \cdot \boldsymbol{c}^{-1})(\boldsymbol{c} \cdot \mathbf{W}^\top)$, which migrates the quantization difficulty from activations to weights, achieving the optimal balance between weight quantization and activation quantization. SmoothQuant enables 8-bit weight-activation quantization without compromising accuracy. To facilitate the quantization of weights, we further integrate GPTQ. We randomly sample 128 text sequences of length 2048 from the self-generated reasoning dataset as the calibration set and leave the other quantization settings the same as QuaRot.

**FlatQuant** (Sun et al., 2024) is the recently proposed approach. Different from QuaRot, it adaptively learns separate Kronecker-decomposed affine transformations to mitigate outliers for each linear layer, i.e., $\mathbf{P}^* = \arg\min_{\mathbf{P}} \|\mathbf{Y} - \mathcal{Q}(\mathbf{XP})\mathcal{Q}(\mathbf{P}^{-1}\mathbf{W}^\top)\|_F^2$. In addition, FlatQuant also introduces learnable clipping and channel-wise scaling to further reduce the quantization error. By default, we use 128 text sequences of length 2048 from Wiki-Text2 (Merity et al., 2016) as the calibration set following Sun et al. (2024). For DeepSeek-R1-Distill-Qwen-1.5B and 7B models, we extend the sequence length to 4096 to mitigate the extreme outlier issues in the K cache as discussed in Appendix C.2. The quantization settings are the same as QuaRot.

**MXFP4** (Rouhani et al., 2023) is part of the Microscaling (MX) data format family, which aims to improve the representation ability of low-precision formats through fine-grained per-group quantization. MXFP4 uses a 4-bit floating-point representation with an E2M1 (2-bit exponent and 1-bit mantissa) format for each element, and a shared 8-bit exponential scaling factor for every group of 32 elements. Similar to QuaRot, we employ a mixed-precision strategy for queries in the self-attention module.

## B Additional Experiments

### B.1 Results on More Quantized LLMs

Below, we present the complete results of QwQ, DeepSeek-R1-Distill-LLaMA, and Qwen3-8B on various reasoning benchmarks.

**Results on QwQ.** From Table 3, it can be seen that conclusions from § 3.1 remain consistent. For example, 4-bit weight-only quantization reaches lossless results while 3-bit induces non-negligible accuracy loss, e.g., over 7% degradation on LiveCodeBench for both AWQ and GPTQ. For KV cache quantization, both KVQuant* and QuaRot achieve lossless 4-bit quantization, but all suffer from severe quantization loss when the KV cache is further quantized into 3 bits. The degradation is most pronounced on difficult tasks with long response

| Model | Quantization | W-A-KV # Bits | Methods | AIME-120 | MATH-500 | GSM8K | GPQA-Diamond | LiveCode-Bench | Avg. | Drop↓ |
|---|---|---|---|---|---|---|---|---|---|---|
| DeepSeek-R1-Distill-LLaMA-8B | **BF16** | - | - | **33.3**±0.8 | **91.0**±1.1 | **88.7**±0.4 | **49.5**±2.3 | **36.6**±3.2 | **59.8**±0.8 | - |
| | *Weight-only* | 4-16-16 | AWQ | 30.3±3.4 | 86.9±0.1 | 87.9±0.7 | 41.1±0.3 | 33.6±2.0 | 56.0±0.7 | -3.9 |
| | | | GPTQ | 30.6±2.1 | 90.0±0.2 | 89.0±0.5 | 47.0±0.5 | 34.6±1.5 | 58.2±0.2 | -1.6 |
| | | 3-16-16 | AWQ | 14.4±1.0 | 77.3±1.4 | 83.5±1.1 | 37.2±1.1 | 22.5±0.2 | 47.0±0.2 | -12.8 |
| | | | GPTQ | 23.3±2.9 | 78.8±0.9 | 73.1±0.6 | 39.1±5.6 | 27.9±1.8 | 48.4±0.6 | -11.4 |
| | *KV Cache* | 16-16-4 | KVQuant* | 36.7±3.8 | 88.5±0.1 | 87.8±0.1 | 45.5±2.2 | 35.5±1.9 | 58.8±1.6 | -1.0 |
| | | | QuaRot | 33.9±2.9 | 90.0±0.7 | 89.1±0.4 | 49.7±1.1 | 36.8±0.9 | 59.9±0.5 | 0.1 |
| | | 16-16-3 | KVQuant* | 19.4±2.7 | 82.0±0.2 | 87.4±0.4 | 41.6±0.8 | 27.1±0.6 | 51.5±0.6 | -8.3 |
| | | | QuaRot | 28.6±4.3 | 87.1±0.2 | 87.9±0.9 | 45.3±3.7 | 31.5±3.0 | 56.1±0.8 | -3.7 |
| | *Weight-Act.* | 8-8-8 | SmoothQuant | 36.7±2.2 | 89.3±0.5 | 88.8±0.5 | 47.8±4.4 | 36.4±1.4 | 59.8±0.3 | 0.0 |
| | | | QuaRot | 38.9±2.9 | 89.9±0.9 | 88.7±0.6 | 48.3±2.8 | 37.7±2.3 | 60.7±0.9 | 0.9 |
| | | | FlatQuant | 35.8±1.4 | 89.0±1.5 | 88.7±0.3 | 48.8±2.5 | 38.1±2.3 | 60.1±1.1 | 0.3 |
| | | 4-4-4 | MXFP4 | 8.9±1.0 | 71.1±1.4 | 81.4±0.4 | 28.5±1.2 | 16.0±0.4 | 41.2±0.3 | -18.7 |
| | | | QuaRot | 16.4±1.0 | 77.3±1.2 | 71.4±1.4 | 32.8±3.2 | 23.3±0.6 | 44.2±1.1 | -15.6 |
| | | | FlatQuant | 21.9±1.3 | 85.5±1.8 | 85.5±0.6 | 42.6±1.8 | 30.0±2.4 | 53.1±1.3 | -6.7 |
| DeepSeek-R1-Distill-LLaMA-70B | **BF16** | - | - | **58.3**±3.8 | **95.6**±0.2 | **94.1**±0.4 | **67.7**±1.8 | **55.9**±0.6 | **74.3**±0.9 | - |
| | *Weight-only* | 4-16-16 | AWQ | 58.3±5.5 | 95.3±0.5 | 93.7±0.3 | 66.3±1.5 | 54.1±0.4 | 73.6±1.2 | -0.8 |
| | | | GPTQ | 59.4±1.3 | 94.4±0.7 | 94.2±0.3 | 63.3±2.5 | 54.6±1.3 | 73.2±0.6 | -1.1 |
| | | 3-16-16 | AWQ | 42.8±2.7 | 93.3±1.2 | 93.5±0.3 | 62.5±3.3 | 48.4±0.9 | 68.1±1.1 | -6.2 |
| | | | GPTQ | 51.4±1.3 | 93.6±0.4 | 94.4±0.2 | 64.0±2.5 | 48.8±2.4 | 70.4±0.8 | -3.9 |
| | *KV Cache* | 16-16-4 | KVQuant* | 55.6±4.6 | 95.1±0.8 | 94.1±0.2 | 69.5±2.4 | 52.5±0.4 | 73.4±1.1 | -1.0 |
| | | | QuaRot | 55.6±1.7 | 95.1±0.4 | 94.2±0.0 | 67.5±2.0 | 54.1±1.5 | 73.3±0.4 | -1.0 |
| | | 16-16-3 | KVQuant* | 51.4±4.6 | 93.3±0.8 | 93.8±0.1 | 63.0±1.1 | 50.1±0.6 | 70.3±1.4 | -4.0 |
| | | | QuaRot | 50.6±1.7 | 94.5±0.3 | 94.2±0.4 | 65.0±2.6 | 52.2±1.3 | 71.3±0.6 | -3.0 |
| | *Weight-Act.* | 8-8-8 | SmoothQuant | 58.1±2.7 | 95.3±0.5 | 94.3±0.4 | 69.2±2.7 | 54.0±1.4 | 74.2±0.7 | -0.1 |
| | | | QuaRot | 59.7±3.4 | 95.3±0.3 | 94.1±0.3 | 65.0±0.8 | 53.7±1.5 | 73.6±0.4 | -0.7 |
| | | | FlatQuant | 62.5±2.5 | 95.7±0.8 | 94.2±0.2 | 66.8±1.8 | 54.9±2.3 | 74.8±0.9 | 0.5 |
| | | 4-4-4 | MXFP4 | 6.1±0.5 | 68.7±0.6 | 88.8±0.3 | 37.9±0.5 | 12.9±0.2 | 42.9±0.1 | -31.4 |
| | | | QuaRot | 9.4±2.7 | 70.0±0.4 | 86.5±0.4 | 33.3±1.8 | 22.1±0.8 | 44.3±0.9 | -30.0 |
| | | | FlatQuant | 54.2±0.8 | 94.7±0.6 | 94.1±0.5 | 66.8±4.1 | 49.5±1.1 | 71.9±0.9 | -2.5 |

Table 4: The overall assessment of quantized DeepSeek-R1-Distill-LLaMA models on various reasoning benchmarks. The green, orange and red cells stand for the lossless (≤1%), the fair (1%-3%) and the risky (≥3%) respectively.

| Model | Quantization | W-A-KV # Bits | Methods | AIME-120 | MATH-500 | GSM8K | GPQA-Diamond | LiveCode-Bench | Avg. | Drop↓ |
|---|---|---|---|---|---|---|---|---|---|---|
| Qwen3-8B | **BF16** | - | - | **68.6**±1.3 | **97.1**±0.6 | **95.3**±0.4 | **60.6**±0.9 | **57.2**±0.4 | **75.8**±0.5 | - |
| | *Weight-only* | 4-16-16 | AWQ | 66.1±1.3 | 97.0±1.0 | 95.0±0.2 | 59.6±0.5 | 54.7±2.1 | 74.5±0.2 | -1.3 |
| | | | GPTQ | 66.9±2.1 | 96.5±0.6 | 95.2±0.1 | 59.4±2.0 | 53.2±0.4 | 74.3±0.6 | -1.5 |
| | | 3-16-16 | AWQ | 44.7±2.4 | 92.9±1.0 | 94.1±0.3 | 46.8±1.5 | 35.3±1.7 | 62.8±0.1 | -13.0 |
| | | | GPTQ | 43.3±2.5 | 92.8±0.7 | 94.1±0.2 | 44.6±1.5 | 31.5±1.6 | 61.3±0.4 | -14.5 |
| | *KV Cache* | 16-16-4 | KVQuant* | 66.7±2.2 | 97.0±0.2 | 95.4±0.2 | 60.9±0.6 | 56.6±0.4 | 75.3±0.5 | -0.5 |
| | | | QuaRot | 70.0±6.0 | 97.3±0.1 | 95.2±0.2 | 59.6±1.0 | 56.7±1.1 | 75.8±1.2 | 0.0 |
| | | 16-16-3 | KVQuant* | 54.4±3.5 | 95.3±0.2 | 94.7±0.3 | 54.0±1.8 | 43.3±0.4 | 68.4±0.6 | -7.4 |
| | | | QuaRot | 57.8±2.7 | 95.7±1.1 | 94.1±0.2 | 50.7±2.4 | 43.5±2.7 | 68.3±0.6 | -7.4 |
| | *Weight-Act.* | 8-8-8 | SmoothQuant | 71.1±4.1 | 96.6±0.4 | 95.3±0.2 | 59.8±1.1 | 56.3±1.4 | 75.8±0.4 | 0.1 |
| | | | QuaRot | 71.1±2.6 | 96.7±0.1 | 95.5±0.1 | 59.6±1.3 | 56.8±0.6 | 76.0±0.3 | 0.2 |
| | | | FlatQuant | 73.6±2.6 | 96.9±0.8 | 95.3±0.3 | 59.3±2.8 | 57.2±1.6 | 76.5±1.4 | 0.7 |
| | | 4-4-4 | QuaRot | 50.0±1.4 | 94.9±0.4 | 94.1±0.1 | 49.5±2.0 | 40.6±1.5 | 65.8±0.8 | -10.0 |
| | | | FlatQuant | 61.1±2.1 | 95.5±0.3 | 94.8±0.2 | 53.7±0.8 | 47.1±0.8 | 70.5±0.5 | -5.3 |

Table 5: The overall assessment of quantized Qwen3-8B model on various reasoning benchmarks. The green, orange and red cells stand for the lossless (≤1%), the fair (1%-3%) and the risky (≥3%) respectively.

lengths such as AIME-120 and LiveCodeBench. For weight-activation quantization, 8-bit quantization remains lossless. For the more challenging 4-bit weight-activation quantization, only FlatQuant maintains the accuracy.

**Results on DeepSeek-R1-Distill-LLaMA.** As shown in Table 4, the findings align with the conclusions presented in § 3.1. LLaMA models can have lossless quantization results for 4-bit KV cache quantization and 8-bit weight-activation quantization. The 4-bit weight quantization is nearly lossless. For lower bits, all of the evaluated quantization methods become risky, e.g. W3G128 incurs accuracy loss over 11% for the 8B model.

**Results on Qwen3-8B.** We further evaluate the performance on the recently released Qwen3-8B. As shown in Table 5, the overall findings remain consistent with our previous analysis in § 3.1. For weight-only quantization, 4-bit methods yield minimal degradation (within 1.5%). However, reducing weights to 3 bits results in substantial performance drops,

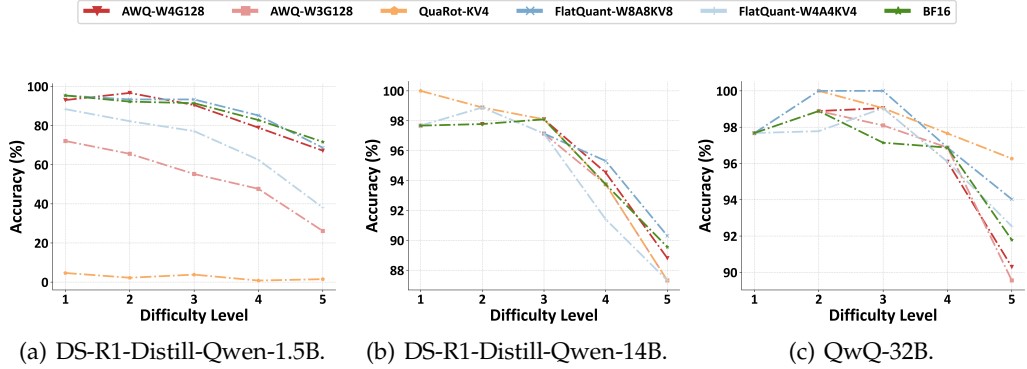

(a) DS-R1-Distill-Qwen-1.5B.  (b) DS-R1-Distill-Qwen-14B.  (c) QwQ-32B.

Figure 6: The accuracy of different difficulty levels on MATH-500.

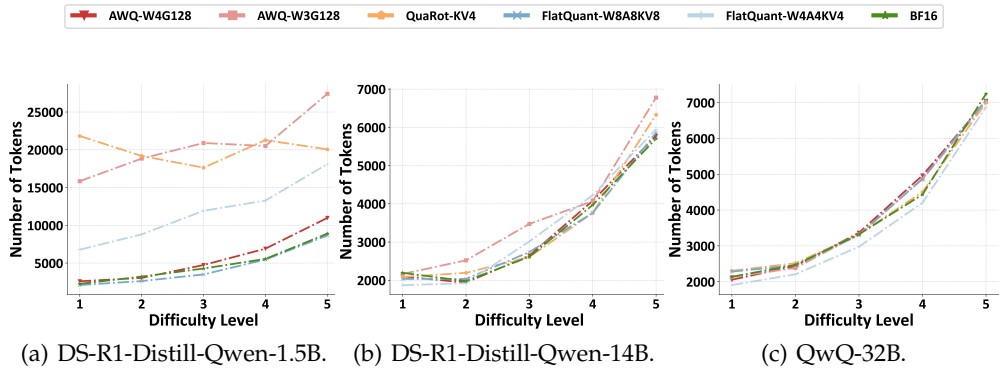

(a) DS-R1-Distill-Qwen-1.5B.  (b) DS-R1-Distill-Qwen-14B.  (c) QwQ-32B.

Figure 7: Number of generated tokens of different difficulty levels on MATH-500.

exceeding 13% on average. For KV cache quantization, both KVQuant* and QuaRot achieve lossless results in 4-bit settings. Yet, when further quantized to 3 bits, they suffer over 7% accuracy loss. Finally, 8-bit weight-activation quantization remains essentially lossless across all evaluated tasks.

## B.2  Performance v.s. Difficulty Levels

Figure 6 presents the performance across varying difficulty levels on the MATH-500 benchmark. The results reveal that quantization-induced accuracy loss becomes more pronounced as task difficulty increases. For example, FlatQuant-W4A4KV4 achieves comparable performance to the full-precision 14B model, while it suffers over 2% accuracy loss at level 5. This indicates that complex tasks exhibit greater vulnerability to precision reduction.

## B.3  Output Length v.s. Difficulty Levels

Here we further examine the impact of task difficulty on the MATH-500 benchmark, which categorizes problems into five difficulty levels. We have the following observations from Figure 7: 1) Reasoning models tend to generate more tokens at test time when solving harder problems. 2) Quantized LLMs with minor accuracy degradation (e.g., W4G128, W8A8KV8) do not generate longer responses than the BF16 model. However, lower bit-width on smaller LLMs (in particular, DeepSeek-R1-Distill-Qwen-1.5B with W3G128 and W4A4KV4 quantization) could lead to increased output, which is similar to the conclusion in § 4.1.

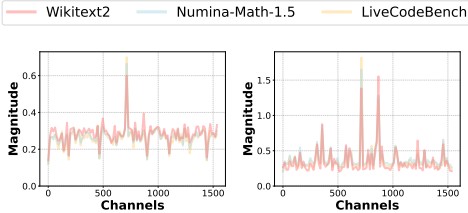 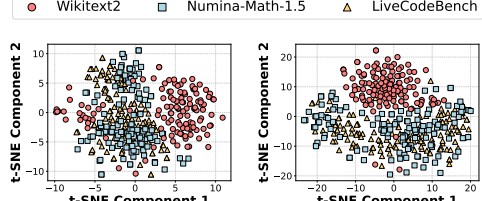

(a) $\mathbf{X}_g$ of the $14^{\text{th}}$ Transformer layer in DeepSeek-R1-Distill-Qwen-1.5B. (b) $\mathbf{X}_q$ of the $21^{\text{st}}$ Transformer layer in DeepSeek-R1-Distill-Qwen-1.5B.

Figure 8: Channel-wise magnitude distributions of activations from different source domains. Reasoning (i.e. Numina-Math-1.5 and LiveCodeBench) and pre-training (i.e. WikiText2) data share similar distributions. The statistics are computed over 32K tokens. $\mathbf{X}_g$ and $\mathbf{X}_q$ denote the inputs of the feed-forward gate projection layer and the self-attention query projection layer in a Transformer layer, respectively.

(a) $\mathbf{X}_g$ of the $14^{\text{th}}$ Transformer layer in DeepSeek-R1-Distill-Qwen-1.5B. (b) $\mathbf{X}_q$ of the $21^{\text{st}}$ Transformer layer in DeepSeek-R1-Distill-Qwen-1.5B.

Figure 9: t-SNE visualization of activations from different source domains. Domain gaps exist between reasoning (i.e. Numina-Math-1.5 and LiveCodeBench) and pre-training (i.e. WikiText2) data. We randomly sample 128 tokens from each domain. $\mathbf{X}_g$ and $\mathbf{X}_q$ denote the inputs of the feed-forward gate projection layer and the self-attention query projection layer in a Transformer layer, respectively.

## C  Additional Discussions

### C.1  Choice of Calibration Data

**Calibration Data Collection.** We collect calibration data from different source domains, including reasoning (i.e. Numina-Math-1.5 (LI et al., 2024) and LiveCodeBench (Jain et al., 2024)) and pre-training (i.e. WikiText2 (Merity et al., 2016)) data. The reasoning data is self-generated by the reasoning model, given the problems in the original dataset, with generation configurations defined in § 3.1.

**The Distribution of Calibration Data from Different Domains.** In Figure 8 and Figure 9, we visualize the channel-wise magnitude and activation distributions from different source domains, respectively. It can be seen that the channel-wise magnitude distributions across different source domains share the same outlier channels, demonstrating notable similarity. Besides, the activation distributions across different reasoning datasets are also close to each other, regardless of the dataset category. However, as shown in Figure 9, activations of the reasoning data exhibit a completely distinct distribution from those of the pre-training data, although their channel-wise magnitude distributions resemble each other. This domain gap may raise issues for post-training quantization methods if pre-training data is used for calibration.

### C.2  Analysis on DeepSeek-R1-Distill-Qwen-1.5B and 7B Models

As discussed in § 3.5, different LLM origins can have various impacts on the quantization of reasoning models. We find that the K cache of DeepSeek-R1-Distill-Qwen-1.5B and 7B models exhibit extreme outlier channels due to the huge bias terms in the self-attention key projection layer, which significantly increases the difficulty of KV cache quantization on these two models. In this section, we first reveal the phenomenon of extreme outlier channels in DeepSeek-R1-Distill-Qwen-1.5B and 7B models as well as its root, and then propose some practical solutions to mitigate its impact on KV cache quantization.

#### C.2.1  Extreme Outlier Channels in K Cache

Qwen (Bai et al., 2023) models add biases in the self-attention query, key, and value projection layers for better length extrapolation ability. However, we find that the bias terms of key

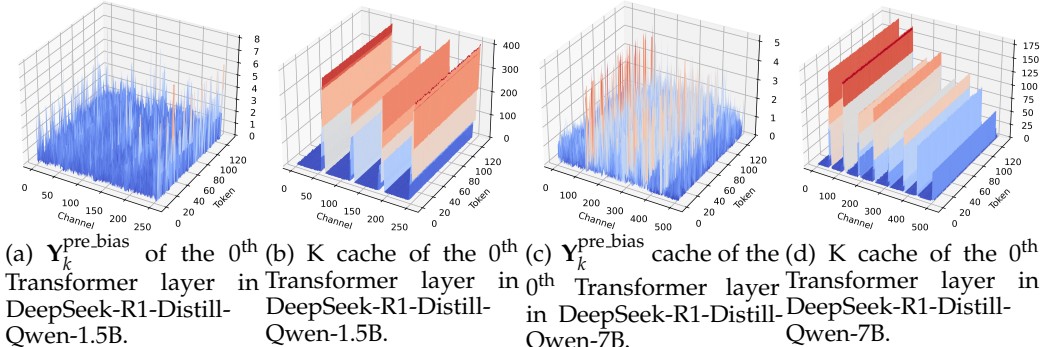

(a) $\mathbf{Y}_k^{\text{pre\_bias}}$ of the $0^{\text{th}}$ Transformer layer in DeepSeek-R1-Distill-Qwen-1.5B.

(b) K cache of the $0^{\text{th}}$ Transformer layer in DeepSeek-R1-Distill-Qwen-1.5B.

(c) $\mathbf{Y}_k^{\text{pre\_bias}}$ cache of the $0^{\text{th}}$ Transformer layer in DeepSeek-R1-Distill-Qwen-7B.

(d) K cache of the $0^{\text{th}}$ Transformer layer in DeepSeek-R1-Distill-Qwen-7B.

Figure 10: Visualizations of K cache before and after the bias term of key projection layers in DeepSeek-R1-Distill-Qwen-1.5B and 7B models. $\mathbf{Y}_k^{\text{pre\_bias}}$ denotes the outputs of the self-attention key projection layer before adding the bias term.

| Model | Methods | W-A-KV # Bits | AIME-120 | MATH-500 | GSM8K | GPQA-Diamond | LiveCode-Bench | Avg. | Drop↓ |
|-------|---------|---------------|----------|----------|-------|--------------|----------------|------|-------|
| **1.5B** | **BF16** | - | **21.7** | **84.4** | **84.6** | **36.9** | **16.0** | **48.7** | - |
| | KVQuant* | 16-16-4 | 7.5 | 58.2 | 67.9 | 29.8 | 10.5 | 34.8 | -13.9 |
| | +Pre_bias | | 20.8 | 83.6 | 84.9 | 31.3 | 16.4 | 47.4 | **-1.3** |
| | KVQuant* | 16-16-3 | 4.2 | 62.4 | 67.2 | 23.7 | 10.5 | 33.6 | -15.1 |
| | +Pre_bias | | 9.2 | 66.2 | 71.3 | 31.3 | 10.5 | 37.7 | **-11.0** |
| **7B** | **BF16** | - | **45.0** | **94.6** | **91.4** | **50.0** | **35.5** | **63.3** | - |
| | KVQuant* | 16-16-4 | 0.0 | 7.0 | 5.0 | 25.3 | 0.4 | 7.5 | -55.8 |
| | +Pre_bias | | 40.8 | 93.2 | 91.4 | 47.5 | 35.1 | 61.6 | **-1.7** |
| | KVQuant* | 16-16-3 | 20.0 | 74.2 | 81.4 | 44.4 | 13.8 | 46.8 | -16.5 |
| | +Pre_bias | | 36.7 | 91.2 | 90.4 | 46.0 | 27.2 | 58.3 | **-5.0** |

Table 6: The effect of pre-bias quantization on DeepSeek-R1-Distill-Qwen-1.5B and 7B models. The green, orange and red cells stand for the lossless ($\leq 1\%$), the fair ($1\%$-$3\%$) and the risky ($\geq 3\%$) respectively.

projection layers can be extremely large in the pre-trained Qwen-1.5B and 7B models, e.g. the maximum absolute value in key projection bias terms reaches 402 in Qwen-1.5B. The distilled reasoning models inherit the large bias terms from pre-trained models, leading to extreme outlier channels on K cache as shown in Figure 10. This phenomenon well explains the large performance gap between KVQuant* and QuaRot on DeepSeek-R1-Distill-Qwen-1.5B and 7B models in Table 1. The presence of extreme outlier channels severely degrades the performance of per-token quantization methods like QuaRot. These outliers force an expansion of the quantization range, leading to catastrophic failures in QuaRot's accuracy. In contrast, KVQuant* mitigates the outlier channels by employing per-channel quantization, which effectively constrains the quantization range and achieves higher accuracy.

### C.2.2 Recipes for Extreme Outlier Channels on K Cache

Based on the observations in § C.2.1, we propose two simple yet effective approaches to improve the KV cache quantization of DeepSeek-R1-Distill-Qwen-1.5B and 7B models. In Table 1, we adopt these approaches as default setups and report their results on DeepSeek-R1-Distill-Qwen-1.5B and 7B models accordingly.

**Pre-bias Per-channel K Cache Quantization for KVQuant*.** As detailed in Appendix A, KVQuant* employs pre-RoPE per-channel K cache quantization. Given that the extreme outlier channels are induced by bias terms, we propose to quantize K cache before adding the bias term in the key projection layer to further reduce the quantization range. As shown in Figure 10, the pre-bias K cache appears much smoother, suggesting greater compatibility with quantization. In Table 6, it can be seen that pre-bias quantization shows consistent

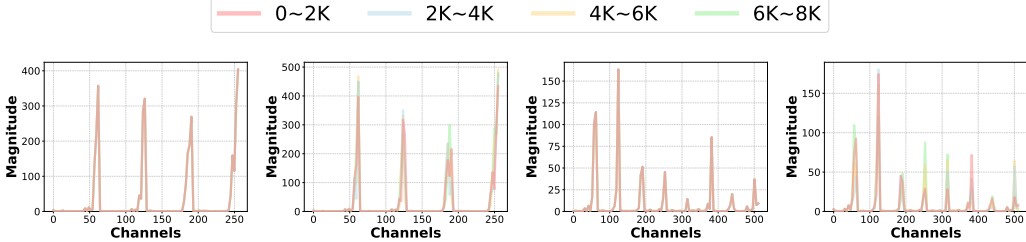

(a) $\mathbf{Y}_k$ of the $0^{th}$ Transformer layer in DeepSeek-R1-Distill-Qwen-1.5B.

(b) K cache of the $0^{th}$ Transformer layer in DeepSeek-R1-Distill-Qwen-1.5B.

(c) $\mathbf{Y}_k$ cache of the $0^{th}$ Transformer layer in DeepSeek-R1-Distill-Qwen-7B.

(d) K cache of the $0^{th}$ Transformer layer in DeepSeek-R1-Distill-Qwen-7B.

Figure 11: Channel-wise magnitude distributions of K cache before and after RoPE. The statistics are computed over different sequence length ranges. $\mathbf{Y}_k$ denotes the outputs of the self-attention key projection layer.

| Methods | Sequence Length | W-A-KV # Bits | Group Size | AIME-120 | MATH-500 | GSM8K | GPQA-Diamond | LiveCode-Bench | Avg. | Drop↓ |
|---------|-----------------|---------------|------------|----------|----------|-------|--------------|----------------|------|-------|
| BF16 | - | - | - | 21.7 | 84.4 | 84.6 | 36.9 | 16.0 | 48.7 | - |
| FlatQuant | 2K | 4-4-4 | N/A | 2.5 | 53.4 | 73.8 | 30.3 | 2.2 | 32.5 | -16.3 |
| | 4K | | | 10.0 | 64.8 | 78.6 | 31.8 | 6.7 | 38.4 | -10.3 |
| | 8K | | | 15.8 | 73.2 | 78.9 | 37.9 | 9.3 | 43.0 | -5.7 |

Table 7: The effect of extending calibration sequence length on DeepSeek-R1-Distill-Qwen-1.5B. The green, orange and red cells stand for the lossless ($\leq$1%), the fair (1%-3%) and the risky ($\geq$3%) respectively.

improvements across different models and bit-widths. For instance, it brings an average accuracy improvement of 7.81% for 3-bit KV cache only quantization.

**Extending the Sequence Length of Calibration Data for FlatQuant.** Reasoning models tend to generate long thinking steps before giving the final answer, which makes it important to maintain the quantization accuracy over very long sequences. However, we find that the extreme outliers induced by bias terms pose a great challenge to learning-based outlier smoothing quantization methods such as FlatQuant, severely degrading the accuracy over long sequences. Specifically, the extreme outliers in the output of the key projection layer are further rotated into adjacent channels through RoPE. As shown in Figure 11, the channel-wise magnitude distributions vary significantly across sequence positions due to differing rotation degrees. As a result, the outlier-smoothing transformations learned on the calibration data with short sequence lengths (e.g. 2K) are unable to generalize to longer sequences. Therefore, we choose to extend the sequence length of calibration data for FlatQuant to mitigate this issue. In Table 7, we vary the lengths of the calibration data in {2K, 4K, 8K}. The accuracy keeps increasing as the length of calibration data increases. In Table 1, we report the results of FlatQuant with the calibration data length of 4K to achieve a balance between training efficiency and quantization accuracy.

# D Qualitative Examples

To learn the concrete behaviors of quantized reasoning models, We present several qualitative examples in Table 8-11.

We divide the reasoning process into four parts — "understand", "process", "check", and "final answer" — and use different colors to represent each part. As shown, the quantized models follow a similar reasoning pattern as the BF16 model, even if the problem is incorrectly answered (Table 11). Nonetheless, the quantized models may occasionally exhibit calculation errors or adopt slightly different verification strategies, but the overall

reasoning structure remains consistent. This suggests that the model's reasoning pattern is relatively robust to quantization, even under aggressive low-bit settings like MXFP4.

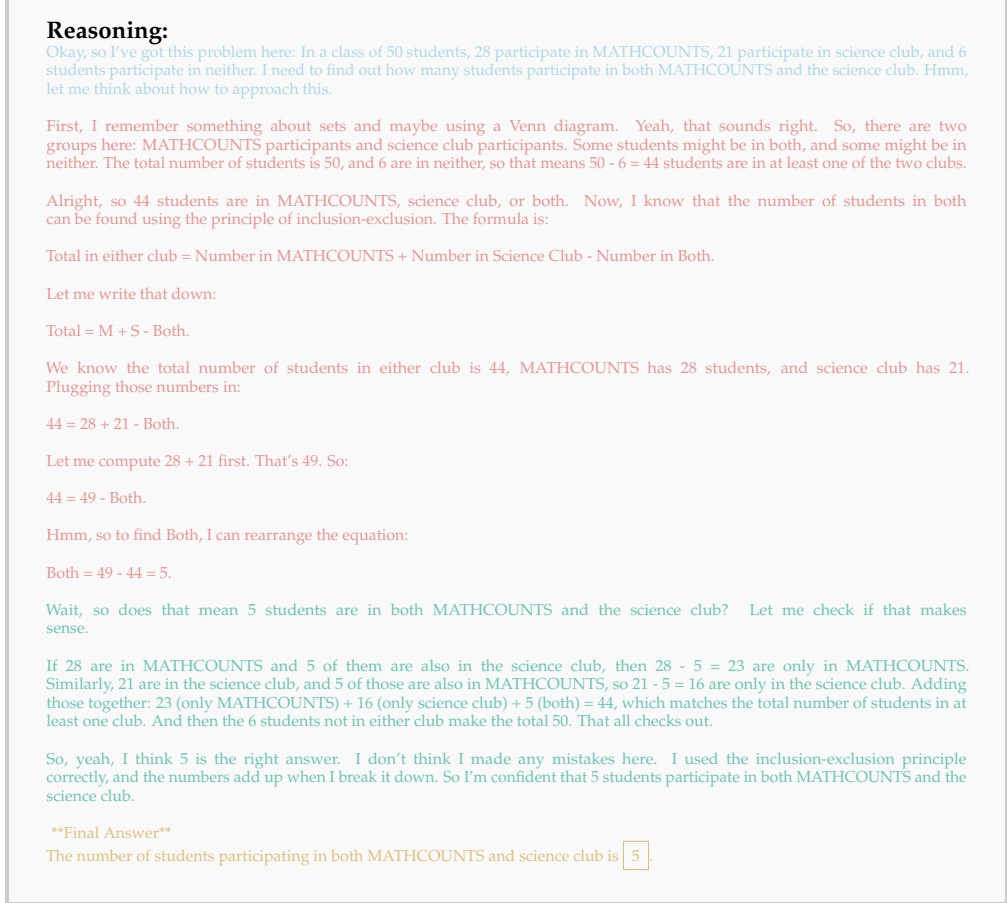

**Problem:** In a class of 50 students, 28 participate in MATHCOUNTS, 21 participate in science club, and 6 students participate in neither. How many students participate in both MATHCOUNTS and science club?

**Reasoning:**

Okay, so I've got this problem here: In a class of 50 students, 28 participate in MATHCOUNTS, 21 participate in science club, and 6 students participate in neither. I need to find out how many students participate in both MATHCOUNTS and the science club. Hmm, let me think about how to approach this.

First, I remember something about sets and maybe using a Venn diagram. Yeah, that sounds right. So, there are two groups here: MATHCOUNTS participants and science club participants. Some students might be in both, and some might be in neither. The total number of students is 50, and 6 are in neither, so that means 50 - 6 = 44 students are in at least one of the two clubs.

Alright, so 44 students are in MATHCOUNTS, science club, or both. Now, I know that the number of students in both can be found using the principle of inclusion-exclusion. The formula is:

Total in either club = Number in MATHCOUNTS + Number in Science Club - Number in Both.

Let me write that down:

Total = M + S - Both.

We know the total number of students in either club is 44, MATHCOUNTS has 28 students, and science club has 21. Plugging those numbers in:

44 = 28 + 21 - Both.

Let me compute 28 + 21 first. That's 49. So:

44 = 49 - Both.

Hmm, so to find Both, I can rearrange the equation:

Both = 49 - 44 = 5.

Wait, so does that mean 5 students are in both MATHCOUNTS and the science club? Let me check if that makes sense.

If 28 are in MATHCOUNTS and 5 of them are also in the science club, then 28 - 5 = 23 are only in MATHCOUNTS. Similarly, 21 are in the science club, and 5 of those are also in MATHCOUNTS, so 21 - 5 = 16 are only in the science club. Adding those together: 23 (only MATHCOUNTS) + 16 (only science club) + 5 (both) = 44, which matches the total number of students in at least one club. And then the 6 students not in either club make the total 50. That all checks out.

So, yeah, I think 5 is the right answer. I don't think I made any mistakes here. I used the inclusion-exclusion principle correctly, and the numbers add up when I break it down. So I'm confident that 5 students participate in both MATHCOUNTS and the science club.

**Final Answer**
The number of students participating in both MATHCOUNTS and science club is 5 .

Table 8: An example from MATH-500 correctly answered by the BF16 DeepSeek-R1-Distill-Qwen-32B model.

**Problem:** In a class of 50 students, 28 participate in MATHCOUNTS, 21 participate in science club, and 6 students participate in neither. How many students participate in both MATHCOUNTS and science club?

**Reasoning:**

Okay, so I've got this problem here: In a class of 50 students, 28 participate in MATHCOUNTS, 21 participate in science club, and 6 students participate in neither. The question is asking how many students participate in both MATHCOUNTS and science club. Hmm, okay, let me think about how to approach this.

First, I remember that when dealing with overlapping groups like this, it's a classic inclusion-exclusion problem. The formula that comes to mind is:

Total = Number in MATHCOUNTS + Number in Science Club - Number in Both + Number in Neither

Wait, let me write that down to make it clearer. So, the total number of students is equal to those in MATHCOUNTS plus those in science club minus those in both, plus those in neither. Hmm, let me make sure I have that right.

Actually, I think the formula is:

Total = (MATHCOUNTS + Science Club - Both) + Neither

Because the ones in neither are outside both groups. So, the total is the union of MATHCOUNTS and Science Club plus the ones in neither. The union is calculated as MATHCOUNTS plus Science Club minus Both.

Let me write that out:

Total = (MATHCOUNTS + Science Club - Both) + Neither

So, plugging in the numbers:

50 = (28 + 21 - Both) + 6

Okay, let me compute that step by step. First, 28 plus 21 is 49. So, 49 minus Both plus 6 equals 50. So, 49 + 6 is 55, right? So, 55 minus Both equals 50.

Hmm, so 55 - Both = 50. Then, subtracting 50 from both sides, we get 55 - 50 = Both, so Both = 5. So, does that mean 5 students are in both MATHCOUNTS and science club?

Wait, let me double-check because sometimes these inclusion-exclusion problems can be tricky. So, if there are 50 students total, and 6 are in neither, that means 50 - 6 = 44 students are in at least one of the two clubs. So, 44 students are in MATHCOUNTS, science club, or both.

Now, MATHCOUNTS has 28 students, and science club has 21. If we add those together, 28 + 21 = 49. But wait, that's more than 44, which is the number of students in at least one club. So, that means the overlap must be 49 - 44 = 5. So, yeah, that confirms it. 5 students are in both.

Let me make sure I'm not making a mistake here. So, if 5 students are in both, then the number only in MATHCOUNTS would be 28 - 5 = 23, and only in science club would be 21 - 5 = 16. Then, adding those up: 23 (only MATHCOUNTS) + 16 (only science) + 5 (both) + 6 (neither) = 23 + 16 is 39, plus 5 is 44, plus 6 is 50. That adds up correctly.

So, yeah, it seems like 5 is the right answer. I think that makes sense because when you have more students in both clubs than the total number in at least one, the difference is the overlap. So, 28 + 21 = 49, but only 44 are in at least one, so 5 must be in both.

Wait, another way to think about it is using a Venn diagram. Imagine two overlapping circles: one for MATHCOUNTS with 28 and one for science club with 21. The total area covered by the circles is 44 students. The sum of the individual circles is 49, so the overlapping part must account for the difference, which is 5. So, yeah, that's consistent.

I guess another way to write the formula is:

Number in Both = MATHCOUNTS + Science Club - Total in at least one

Which would be:

Number in Both = 28 + 21 - 44 = 5

So, that also gives 5. So, all the methods point to 5. I think I'm confident that 5 is the correct answer.

Just to recap:

- Total students: 50
- Neither: 6
- Therefore, in at least one club: 50 - 6 = 44
- MATHCOUNTS: 28
- Science Club: 21
- Total in both clubs: 28 + 21 = 49
- But since only 44 are in at least one, the overlap is 49 - 44 = 5

Yeah, that seems solid. So, I think the answer is 5 students participate in both MATHCOUNTS and science club.

**Final Answer**
The number of students who participate in both MATHCOUNTS and science club is $\boxed{5}$.

Table 9: An example from MATH-500 correctly answered by the DeepSeek-R1-Distill-Qwen-32B model quantized with GPTQ-W4A16G128

**Problem:** Let $A = (1, -11, 2)$, $B = (3, -4, 1)$, and $C = (-2, 1, -1)$. Compute $\angle ABC$, in degrees.

**Reasoning:**

Okay, so I need to find the angle at point B in triangle ABC, which is angle ABC. The points are given as A = (1, -11, 2), B = (3, -4, 1), and C = (-2, 1, -1). Hmm, I remember that to find the angle between two vectors, I can use the dot product formula. Since angle ABC is the angle at point B, I think I need to find the vectors BA and BC, and then find the angle between them.

Let me first visualize this. Point B is the vertex where the angle is, so BA is the vector from B to A, and BC is the vector from B to C. Once I have these two vectors, I can use the dot product formula to find the angle between them.

So, how do I find vectors BA and BC? Well, vectors can be found by subtracting the coordinates of the initial point from the terminal point. So for vector BA, it's A minus B, and for vector BC, it's C minus B. Let me write that down.

Vector BA = A - B = (1 - 3, -11 - (-4), 2 - 1) = (-2, -7, 1)
Vector BC = C - B = (-2 - 3, 1 - (-4), -1 - 1) = (-5, 5, -2)

Okay, so now I have BA = (-2, -7, 1) and BC = (-5, 5, -2). Next step is to find the dot product of BA and BC, and then divide it by the product of their magnitudes. That should give me the cosine of the angle between them, right?

Let me compute the dot product first. The dot product of BA and BC is calculated by multiplying corresponding components and adding them up. So:

$BA \cdot BC = (-2)(-5) + (-7)(5) + (1)(-2) = 10 - 35 - 2 = -27$

Hmm, that's a negative number. That means the angle is obtuse, since the cosine is negative. Okay, moving on.

Now, I need the magnitudes of BA and BC. The magnitude of a vector is found by taking the square root of the sum of the squares of its components.

First, magnitude of BA:

$|BA| = sqrt[(-2)^2 + (-7)^2 + (1)^2] = sqrt[4 + 49 + 1] = sqrt[54] = 3 * sqrt(6)$

Wait, sqrt(54) is equal to sqrt(9*6) which is 3*sqrt(6). That seems right.

Now, magnitude of BC:

$|BC| = sqrt[(-5)^2 + (5)^2 + (-2)^2] = sqrt[25 + 25 + 4] = sqrt[54] = 3 * sqrt(6)$

Oh, interesting, both vectors BA and BC have the same magnitude, 3*sqrt(6). So that simplifies things a bit.

Now, using the dot product formula:

$cos(theta) = (BA \cdot BC)/(|BA||BC|)$

Plugging in the numbers:

$cos(theta) = (-27)/(3 * sqrt(6) * 3 * sqrt(6)) = (-27)/(9 * 6) = (-27)/54 = -0.5$

So, cos(theta) = -0.5. Now, to find theta, I need to take the arccosine of -0.5.

I remember that cos(120°) = -0.5, because cos(60°) is 0.5 and cosine is negative in the second quadrant. So, arccos(-0.5) is 120 degrees.

Therefore, angle ABC is 120 degrees.

Wait, just to double-check everything. Let me recap:

1. Vectors BA and BC are correct. BA is (-2, -7, 1), BC is (-5, 5, -2). Yes, that's right because we subtract B from A and B from C.

2. Dot product: (-2)(-5) = 10, (-7)(5) = -35, (1)(-2) = -2. So 10 - 35 -2 = -27. That's correct.

3. Magnitudes: sqrt(4 + 49 +1) = sqrt(54), and sqrt(25 +25 +4) = sqrt(54). So both are 3*sqrt(6). Correct.

4. So, cos(theta) = (-27)/(54) = -0.5. Yes, that's right.

5. Arccos(-0.5) is 120 degrees. That's correct because in the unit circle, 120 degrees is in the second quadrant with cosine -0.5.

Therefore, I think I did everything correctly, and the angle is indeed 120 degrees.

**Final Answer**
The measure of angle ABC is $\boxed{120}$ degrees.

Table 10: An example from MATH-500 correctly answered by the BF16 DeepSeek-R1-Distill-Qwen-32B model.

**Problem:** Let $A = (1, -11, 2)$, $B = (3, -4, 1)$, and $C = (-2, 1, -1)$. Compute $\angle ABC$, in degrees.

**Reasoning:**

Okay, so I need to find the angle at point B in triangle ABC. The points given are A = (1, -11, 2), B = (3, -4, 1), and C = (-2, 1, -1). Wait, hold on, the coordinates for point C are written as (-, 1, -1). That doesn't look right. Is that a typo or a mistake? Maybe it should be something like (-1, 1, -1) or (-2, 1, -1) with a specific number? Hmm, maybe it's a typo and supposed to be (-1, 1, -1). I'll assume that for now, so point C is (-1, 1, -1). I think that makes sense.
So, points are A = (1, -11, 2), B = (3, -4, 1), and C = (-1, 1, -1). I need to compute angle ABC, which is the angle at point B between points A, B, and C. So, angle ABC is the angle at vertex B, formed by sides BA and BC. So, to find this angle, I need to find the vectors BA and BC, then compute the angle between these two vectors using the dot product formula.

First, I need to compute vectors BA and BC.
Vector BA is from B to A, so it's A - B. Similarly, vector BC is from B to C, so it's C - B.
Let me compute vector BA first.
A is (1, -11, 2), B is (3, -4, 1). So vector BA is A - B, which is (1 - 3, -11 - (-4), 2 - 1). Let me calculate each component:
1 - 3 is -2.
-11 - (-4) is -11 + 4, which is -7.
2 - 1 is 1.
So vector BA is (-2, -7, 1). Wait, hold on, that would be (-2, -7, 1). Wait, is that right? 1 - 3 is -2, so first component is -2. Second component is -11 - (-4) is -11 + 4 is -7. Third component is 2 - 1 is 1. So vector BA is (-2, -7, 1). Wait, maybe the first component is -2, so it's (-2, -7, 1) or (-2, -7, 1). Hmm, I think it's (-2, -7, 1). Maybe the person meant to write (-2, -7, 1), but the first component is -2. So, in components, it's (-2, -7, 1). So, BA = (-2, -7, 1). Wait, maybe it's (-2, -7, 1), but the first component is -2.
...

Therefore, arccos(-0.588) is 180 - 54 = 126 degrees.

Wait, 54 degrees, so 180 - 54 is 126 degrees. So, the angle is approximately 126 degrees.

Wait, let me confirm.

If arccos(-0.588) is approximately 126 degrees because cos(126 degrees) is $cos(180 - 54) = -cos(54) \approx -0.5878$, which is about -0.588. So, that's exactly what we have. So, $cos(theta) \approx -0.588$, so theta is approximately 126 degrees.

So, the angle ABC is approximately 126 degrees.

Wait, let me double-check my calculations.

First, vectors BA and BC:

Vector $BA : A - B = (1 - 3, -11 - (-4), 2 - 1) = (-2, -7, 1)$

$Vector BC : C - B = (-1 - 3, 1 - (-4), -1 - 1) = (-4, 5, -2)$

...

Vector BC is from B to C, which is $C - B$, which is $(-1 - 3, 1 - (-4), -1 - 1) = (-4, 5, -2)$. Correct.

Dot product: $(-2) * (-4) = 8, (-7) * (5) = -35, (1) * (-2) = -2$. Total: $8 - 35 - 2 = -29$. Correct.

Magnitudes: $|BA| = sqrt[(-2)^2 + (-7)^2 + 1^2] = sqrt[4 + 49 + 1] = sqrt[54] = 3 * sqrt(6)$. Correct.

$|BC| = sqrt[(-4)^2 + 5^2 + (-2)^2] = sqrt[16 + 25 + 4] = sqrt[45] = 3 * sqrt(5)$. Correct.

So, $cos(theta) = (-29)/(3 * sqrt(6) * 3 * sqrt(5)) = (-29)/(9 * sqrt(30))$.

Yes, that's correct. So, the angle is arccos(-29/(9*sqrt(30))).

So, in degrees, that is approximately 126 degrees.

Therefore, the angle ABC is approximately 126 degrees.

I think that's the answer.

*Final Answer**
The measure of angle ABC is 126 degrees.

Table 11: An example from MATH-500 that was incorrectly answered by the MXFP4-quantized DeepSeek-R1-Distill-Qwen-32B model.

