# OpenReview forum: "Quantization Hurts Reasoning? An Empirical Study on Quantized Reasoning Models"
_colmweb.org/COLM/2025/Conference — COLM 2025_

### Official Review · Reviewer_xjtt · 2025-05-10

**Rating:** 9
**Confidence:** 4
**Ethics Flag:** 1

**Summary:**

This paper presents an empirical study on quantization in reasoning models and finds that lossless quantization can be achieved with W8A8 or W4A16 quantization; while, lower bit-widths introduce significant accuracy risks. The paper points to model size, model origin, and task difficulty as critical determinants of performance. The paper contends that strategically scaling model sizes or reasoning steps can enhance performance.

**Reasons To Accept:**

Code will be made open source.

I also like the fact that the findings are summarized in the section that is usually used to detail “contributions”, and these findings are further summarized (paraphrasing and quoting) as follows:
1. Lossless Quantization is possible:
8-bit weight-activation quantization preserves accuracy across tasks and model sizes, while
4-bit weight-only quantization or KV cache quantization also achieves near-lossless results (
2. Algorithm Guidance:
The paper suggests the following methods are top performers:
* AWQ for weight-only quantization and
* QuaRot for KV cache quantization and
* FlatQuant for weight-activation quantization.
3. Task Difficulty Impact:
Harder tasks (e.g., AIME-120) suffer up to 4×greater degradation than simpler ones (e.g., GSM8K).
4. Distillation vs not: Distillation-based reasoning models withstand quantization better than RL-based counterparts (e.g., QwQ), while Qwen and LLaMA families exhibit different tolerance to quantization.
5. Analysis of Output Length: Quantized models with minor performance drops do not exhibit longer reasoning steps, but aggressive low-bit quantization can lead to increased output lengths, particularly in smaller models.
6. Scaling Effects: Larger quantized models achieve superior accuracy latency trade-offs compared to smaller BF16 models, and while longer reasoning.

Table 1 results looks quite extensive.

**Reasons To Reject:**

None. Good paper.

---

> ### Author Response · Authors · 2025-06-03
>
> Thanks for your positive reviews! We sincerely appreciate the time and effort you have dedicated to evaluating our work.

---

> > ### Comment · Reviewer_xjtt · 2025-06-06
> > **Holding with my strong positive review**
> >
> > I have read the other reviews and responses. I continue to like the useful information that this paper provides to the community and I will continue to champion it unless another reviewer can provide some strong outstanding reason for me to reduce my level of positivity. I feel the responses to the other reviewers made me feel less concerned about the other reviewers reasons to not accept this paper. I am not concerned about the level of novelty issue. This paper provides value.

---

> > > ### Author Response · Authors · 2025-06-09
> > >
> > > Dear Reviewer xjtt,
> > >
> > > Many thanks for your recognition and support for our work!
> > >
> > > Best,
> > > Authors

---

### Official Review · Reviewer_QCfM · 2025-05-11

**Rating:** 8
**Confidence:** 3
**Ethics Flag:** 1

**Summary:**

This paper presents a systematic analysis on the impact of the quantization of reasoning models.
They experiment on a wide range of RMs and quantization methods.
The results provide many insights on understanding quantization of RMs.

**Questions To Authors:**

1. Is there any hypothesis why more difficult reasoning tasks and more reasoning steps show greater performance decline with quantization? Does precision of computation relate to the performance of reasoning?

**Reasons To Accept:**

1. The paper is well-written and easy for readers to follow. The conclusions are highlighted.
2. The analysis is thorough, including many main-stream quantization methods and a wide range of RMs.
3. The study offers practical takeaways, including recommending specific quantization algorithms for different purposes.

**Reasons To Reject:**

Not reasons to reject, but some suggestions for improvement. An in-depth analysis into the mechanism of quantization which leads to performance degradation will improve the paper further.

---

> ### Author Response · Authors · 2025-06-03
>
> **Q1.** *An in-depth analysis into the mechanism of quantization which leads to performance degradation will improve the paper further.*
>
>
> **A1**. Thanks for your valuable suggestion. More in-depth analysis can be found in appendix due to limited space. For example:
>
> 1. **Calibration Data Matters**. 1) **GPTQ**. We find that using reasoning data for calibration is critical for quantization methods that rely heavily on the calibration data for quantization error compensation (e.g. GPTQ). 2) **AWQ**. On the other hand, methods that only rely on calibration data for quantization parameter computation or outlier channel identification are robust to calibration data selection (e.g. AWQ). Detailed explanations are provided in Appendix C.1.
>
> 2. **Be Careful of Model-specific Features**. As shown in Table 1, we find that QuaRot generally outperforms KVQuant* thanks to the Hadamard transformation. However, as detailed in Appendix C.2, for models with extreme outlier channels on K cache, KVQuant* can significantly outperform QuaRot through per-channel K cache quantization, and the performance can be further boosted with the pre-bias quantization introduced in L628-645. These results show that quantization accuracy can be significantly improved by targeting model-specific features.
>
>
> ---
> **Q2**. *Is there any hypothesis why more difficult reasoning tasks and more reasoning steps show greater performance decline with quantization? Does precision of computation relate to the performance of reasoning?*
>
>
> **A2**. Thanks for the question.
>
>
> As shown in Figure 5, the test-time scaling of low-bit quantized reasoning models is inferior to their full-precision counterparts. In L304-305, we speculate that the quantization error accumulated along the sequence may hinder effective reasoning, which makes the accuracy gap grow larger as the reasoning steps increase.
>
>
> At a higher level, we hypothesize that quantization may impair the knowledge encoded in model weights as well as the reasoning ability, which combine to hurt the performance of quantized models on hard problems. We plan to systematically evaluate the knowledge loss and reasoning ability loss brought by quantization through the lens of Pass@K and Pass@1 metrics in future work.

---

> > ### Comment · Reviewer_QCfM · 2025-06-09
> >
> > I think the authors have addressed all my concerns. I will maintain my score.

---

### Official Review · Reviewer_kP3E · 2025-05-11

**Rating:** 6
**Confidence:** 3
**Ethics Flag:** 1

**Summary:**

This paper provides a comprehensive empirical study on the quantization of reasoning models. They evaluate SoTA quantization methods across weight-only, weight-activation, and KV cache quantization. Their conclusion is:
Lossless settings: W8A8 or W4A16 are consistently < 1 % accuracy drop; W4 weight‑only or KV‑cache is “near‑lossless” on most models.
Best algorithms: AWQ for weight‑only, QuaRot for KV‑cache (except small Qwen), FlatQuant for weight‑activation.
Failure modes: 3‑bit quantization, small models, harder tasks and RL‑trained models (QwQ) are much more fragile.
No extra thinking: mild quantization does not lengthen CoT while aggressive low‑bit settings do.
Scaling trade‑offs: 4‑bit large models beat smaller BF16 models in both accuracy‑per‑GB and accuracy‑per‑second.

**Reasons To Accept:**

Comprehensive and solid experiments focus on everything in quantization.

**Reasons To Reject:**

1. No novel ideas are proposed. It is more about empirical study paper. The main contribution of this paper is doing experiments for the current quantization algorithms, and conclusion might be change as new approaches evolve. This paper would be benefit from this by proposing a leaderboard that support evaluations of any quantization algorithms.
2. It does not really explain why certain quantization algorithms fail or do well. For example, it states "AWQ is preferred over GPTQ", "QuaRot is generally preferred over KVQuant*". This paper could be benefit if author can connect those empirical conclusions (accuracy drop) with some theoretically justifications and provide some insights in designing new algorithms.

---

> ### Author Response · Authors · 2025-06-03
>
> **Q1.** *No novel ideas are proposed. It is more about empirical study paper. The main contribution of this paper is doing experiments for the current quantization algorithms, and conclusion might be change as new approaches evolve. This paper would be benefit from this by proposing a leaderboard that support evaluations of any quantization algorithms.*
>
>
> **A1**. Thanks for the suggestion. We fully understand your concerns as new approaches are evolving rapidly nowadays. On the one hand, we do our best to provide **findings generalizable to different models**. On the other hand, to continuously help the community, we will **release our code** ([https://anonymous.4open.science/r/Quantized-Reasoning-Models-Anonymous-419C/README.md](https://anonymous.4open.science/r/Quantized-Reasoning-Models-Anonymous-419C/README.md)), which is built upon vLLM and LightEval that allows easy extension to more recent reasoning models. The idea to maintain a live leaderboard aligns with our plans, and we have started to prepare this already.
>
>
> For example, we integrate it with a recently released reasoning model Qwen3-8B, and the results align with the general findings in Table 1 of our paper. For instance, 8-bit weight and activation is still lossless on most benchmarks for Qwen3-8B.
>
> |Quantized Model (Acc)|AIME-120|MATH-500|GSM8K|GPQA-Diamond|LiveCodeBench|Avg.|
> |:----|:----|:----|:----|:----|:----|:----|
> |Qwen3-8B|68.61|97.07|95.32|60.61|57.21|75.76|
> |AWQ-w4g128|66.11|97.00|94.97|59.60|54.73|74.48|
> |GPTQ-w4g128|66.94|96.53|95.15|59.43|53.23|74.26|
> |AWQ-w3g128|44.72|92.93|94.09|46.80|35.32|62.77|
> |GPTQ-w3g128|43.33|92.80|94.11|44.61|31.47|61.27|
> |KVQuant*-kv4|66.67|97.00|95.35|60.94|56.59|75.31|
> |QuaRot-kv4|70.00|97.27|95.22|59.60|56.72|75.76|
> |KVQuant*-kv3|54.44|95.27|94.74|54.04|43.28|68.36|
> |QuaRot-kv3|57.78|95.67|94.06|50.67|43.53|68.34|
> |SmoothQuant-w8a8kv8|71.11|96.60|95.25|59.76|56.34|75.81|
> |QuaRot-w8a8kv8|71.11|96.73|95.50|59.60|56.84|75.96|
> |FlatQuant-w8a8kv8|73.61|96.93|95.30|59.26|57.21|76.46|
> |QuaRot-w4a4kv4|50.00|94.93|94.09|49.49|40.55|65.81|
> |FlatQuant-w4a4kv4|61.11|95.53|94.79|53.70|47.14|70.46|
>
>
> ---
> **Q2.** *It does not really explain why certain quantization algorithms fail or do well. For example, it states "AWQ is preferred over GPTQ", "QuaRot is generally preferred over KVQuant*". This paper could be benefit if author can connect those empirical conclusions (accuracy drop) with some theoretically justifications and provide some insights in designing new algorithms.*
>
> **A2**. We have analyzed why certain algorithms are better than others in Appendix C and will move this part to the main text in the revision.
>
> 1. **Calibration Data Matters**. 1) **GPTQ**. We find that using reasoning data for calibration is critical for quantization methods that rely heavily on the calibration data for quantization error compensation (e.g. GPTQ). 2) **AWQ**. On the other hand, methods that only rely on calibration data for quantization parameter computation or outlier channel identification are robust to calibration data selection (e.g. AWQ). Detailed explanations are provided in Appendix C.1.
>
> 2. **Be Careful of Model-specific Features**. As shown in Table 1, we find that QuaRot generally outperforms KVQuant* thanks to the Hadamard transformation. However, as detailed in Appendix C.2, for models with extreme outlier channels on K cache, KVQuant* can significantly outperform QuaRot through per-channel K cache quantization, and the performance can be further boosted with the pre-bias quantization introduced in L628-645. These results show that quantization accuracy can be significantly improved by targeting model-specific features.

---

> > ### Comment · Reviewer_kP3E · 2025-06-09
> >
> > This paper has done a significant amount of work in the rebuttal and has addressed my concerns; thus, I have raised the score.

---

> > > ### Author Response · Authors · 2025-06-09
> > >
> > > Dear Reviewer kP3E,
> > >
> > > Appreciation for raising your score and many thanks again for your recognition!
> > >
> > > Best,
> > > Authors

---

### Official Review · Reviewer_v9cz · 2025-05-12

**Rating:** 6
**Confidence:** 4
**Ethics Flag:** 1

**Summary:**

The paper presents a comprehensive empirical study of quantization methods on a series of reasoning models, highlighting the strengths / weaknesses of the corresponding quantization techniques and approaches, as well as general performance of the quantized systems when it comes to task accuracy (and dropoff characteristics) and output length.

**Questions To Authors:**

None.

UPDATE: Score raised to 6 following rebuttal period discussions.

**Reasons To Accept:**

- An empirical study of quantization models in the reasoning model context is a valuable contribution that can help guide practitioners.
- The experiments are comprehensive and cover models that would commonly be applied in most contexts.

**Reasons To Reject:**

- The empirical contributions of this paper, though useful, do not yield any novel insights or clearly generalizable advice / rules of thumb that can be applied more generally in the field. While the paper does discuss preferred algorithms for quantization, it is not clear (and plausibly not correct) that these optimal algorithms would remain optimal, for, say, the next generation of reasoning models. Moreover, there is no conceptual analysis explaining or interpreting the findings and leading to higher-level findings. As a result, the study feels like a snapshot of what works now, as opposed to what we can learn about quantization in this space of models more generally.

- Building on the previous point, I'm concerned that this paper will quickly lose relevance as the landscape continues to evolve rapidly: In an ideal world, this paper would be better shielded from this by proposing a framework / codebase that automates this quantization evaluation and maintains a live leaderboard. However, I don't see any such work proposed here.

All in all, I find that investigating the effects of quantization on large reasoning models is a valuable, if straightforward, research goal. However, the paper merely focuses on this task for the time being, and does not provide tools / lessons that can apply going forward, which seriously hinders its significance in my opinion.

---

> ### Author Response · Authors · 2025-06-03
>
> **Q1.** *The empirical contributions of this paper, though useful, do not yield any novel insights or clearly generalizable advice / rules of thumb that can be applied more generally in the field. While the paper does discuss preferred algorithms for quantization, it is not clear (and plausibly not correct) that these optimal algorithms would remain optimal, for, say, the next generation of reasoning models. Moreover, there is no conceptual analysis explaining or interpreting the findings and leading to higher-level findings. As a result, the study feels like a snapshot of what works now, as opposed to what we can learn about quantization in this space of models more generally.*
>
>
> **A1**. **We conduct extensive experiments with the best effort to obtain generalizable conclusions, as outlined L48 - L62**. To verify this, we further conduct more experiments on the recently released reasoning model Qwen3-8B, with results listed in A2. Our findings still apply to this new model, e.g., the lossless quantization can be achieved with 8-bit weight and activation quantization, etc.
>
>
> As the first study in this field, we seek to balance the width and depth of this research. Aside from the general guidelines, we have also provided more conceptual analysis to strengthen the understanding of quantized reasoning models in our paper:
>
> * **Insights into the Performance of Quantization Algorithms**. In Appendix C, we provide discussions on the underlying reason why certain quantization algorithms are preferred for reasoning models. For example, we find that the choice of calibration data matters for quantization methods that rely heavily on the calibration data for quantization error compensation (e.g. GPTQ) with reasons explained in Appendix C.1.
>
> * **Test-time Scaling of Quantized Reasoning Models**. We first explore the test-time scaling of quantized reasoning models in Section 4.3, and find that the test-time scaling of low-bit quantized models is inferior to that of BF16 models, which may partially explain why harder problems with longer answers suffer more from quantization.
>
> * **Effectiveness on Real-world Deployment**. In Sections 4.1 & 4.2, we confirm that quantized reasoning models can effectively benefit inference latency by decreasing step-wise latency and maintaining the output length.
>
>
> ---
> **Q2.** *Building on the previous point, I'm concerned that this paper will quickly lose relevance as the landscape continues to evolve rapidly: In an ideal world, this paper would be better shielded from this by proposing a framework / codebase that automates this quantization evaluation and maintains a live leaderboard. However, I don't see any such work proposed here.*
>
>
> **A2**. Thanks for your insightful advice. We completely understand your concerns, especially given how quickly LLMs are evolving these days. To continuously help the community, we will release our code ([https://anonymous.4open.science/r/Quantized-Reasoning-Models-Anonymous-419C/README.md](https://anonymous.4open.science/r/Quantized-Reasoning-Models-Anonymous-419C/README.md)), which is built upon vLLM and LightEval that allows easy extension to more recent reasoning models.
>
>
> For example, we integrate it with a recently released reasoning model Qwen3-8B, and the results align with the general findings in Table 1 of our paper. For instance, 8-bit weight and activation is still lossless on most benchmarks for Qwen3-8B.
>
> |Quantized Model (Acc)|AIME-120|MATH-500|GSM8K|GPQA-Diamond|LiveCodeBench|Avg.|
> |:----|:----|:----|:----|:----|:----|:----|
> |Qwen3-8B|68.61|97.07|95.32|60.61|57.21|75.76|
> |AWQ-w4g128|66.11|97.00|94.97|59.60|54.73|74.48|
> |GPTQ-w4g128|66.94|96.53|95.15|59.43|53.23|74.26|
> |AWQ-w3g128|44.72|92.93|94.09|46.80|35.32|62.77|
> |GPTQ-w3g128|43.33|92.80|94.11|44.61|31.47|61.27|
> |KVQuant*-kv4|66.67|97.00|95.35|60.94|56.59|75.31|
> |QuaRot-kv4|70.00|97.27|95.22|59.60|56.72|75.76|
> |KVQuant*-kv3|54.44|95.27|94.74|54.04|43.28|68.36|
> |QuaRot-kv3|57.78|95.67|94.06|50.67|43.53|68.34|
> |SmoothQuant-w8a8kv8|71.11|96.60|95.25|59.76|56.34|75.81|
> |QuaRot-w8a8kv8|71.11|96.73|95.50|59.60|56.84|75.96|
> |FlatQuant-w8a8kv8|73.61|96.93|95.30|59.26|57.21|76.46|
> |QuaRot-w4a4kv4|50.00|94.93|94.09|49.49|40.55|65.81|
> |FlatQuant-w4a4kv4|61.11|95.53|94.79|53.70|47.14|70.46|
>
> BTW, the idea to maintain a live leaderboard aligns with our plans, and we have started to prepare this already. Thanks for the wonderful suggestion :)

---

> > ### Comment · Reviewer_v9cz · 2025-06-08
> > **Reviewer Response**
> >
> > I thank the authors for their response, and am happy to see that they are considering this direction going forward. I'm also happy to see the discussions in the Appendix, which I recommend to be included in the paper itself. As such, I will increase my score.

---

> > ### Author Response · Authors · 2025-06-09
> >
> > Dear Reviewer v9cz,
> >
> > Thanks for raising your evaluation and valuable suggestions. As there is one additional page available in the final version, we will revise our manuscript accordingly, together with some up-to-date results (e.g., Qwen 3 and more testing results).
> >
> > Best,
> > Authors

---

### Author Response · Authors · 2025-06-03
**The General Response**

We sincerely thank all reviewers for their insightful comments and suggestions. We appreciate the recognition of this work, including the practicality (v9cz, QCfM), comprehensiveness (v9cz, kP3E, QCfM, xjtt) and solid experiments (kP3E).


We would like to highlight that our work is the first comprehensive evaluation of the existing quantization methods on reasoning models. Below is a summary of our responses and main contributions:

* **Practical Guidelines on Quantization of Reasoning Models**. We conduct extensive evaluations and summarize key findings to facilitate the real-world deployment of quantized reasoning models, such as the lossless quantization settings and the preferred quantization methods. As noted in A1 of Reviewer v9cz, these findings can be generalized to the latest models, such as Qwen3 as well.

* **In-depth Analysis**. As the first study, we try to keep the balance between the width (the general guidelines) and depth (the underlying mechanisms of quantized reasoning models) in this research (Reviewer v9cz and Reviewer kP3E). For the latter, we study 1) the test-time scaling of quantized reasoning models (Sections 4.2 & 4.3); 2) the effect of calibration data (Appendix C.1); and 3) the characteristics of specific models (Appendix C.2), etc. We hope these discussions can help strengthen the understanding of quantized reasoning models.

* **Open-sourced Codebase and Live Leaderboard for Continual Update**. We will release our codebase ([https://anonymous.4open.science/r/Quantized-Reasoning-Models-Anonymous-419C/README.md](https://anonymous.4open.science/r/Quantized-Reasoning-Models-Anonymous-419C/README.md)) of this work, which is built upon vLLM and LightEval that allows easy extension to more recent reasoning models (e.g. Qwen 3 as listed in A2 of Reviewer v9cz), quantization methods, and tasks. Meanwhile, we will prepare to maintain a live leaderboard of quantized reasoning models, and thanks for the suggestion from Reviewer v9cz and Reviewer kP3E.


In summary, we hope our research could help the community pay more attention to the quantization (and compression) of reasoning models. We will continually update our research (e.g., codebase & leaderboard) so as to provide a feasible platform for the community to conduct further research in this area.

---

### Decision · Program_Chairs · 2025-07-08

**Decision:**

Accept

**Comment:**

This paper provides a comprehensive empirical study of the effect of quantization on the reasoning capabilities of LLMs. Aspects studied include quantization method and amount, task difficulty, and scaling in both length and model size. Interesting findings were proposed with clear takeaways. After discussion, all reviewers are positive about the work and recommend acceptance.